# *MaGNET*: Uniform Sampling from Deep Generative Network Manifolds Without Retraining

**Ahmed Imtiaz Humayun**
Rice University
imtiaz@rice.edu

**Randall Balestriero**
Rice University
randallbalestriero@gmail.com

**Richard Baraniuk**
Rice University
richb@rice.edu

## ABSTRACT

Deep Generative Networks (DGNs) are extensively employed in Generative Adversarial Networks (GANs), Variational Autoencoders (VAEs), and their variants to approximate the data manifold and distribution. However, training samples are often distributed non-uniformly on the manifold, due to the cost or convenience of collection. For example, the CelebA dataset contains a large fraction of smiling faces. *These inconsistencies will be reproduced when sampling from the trained DGN, which is not always preferred, e.g., for fairness or data augmentation.* In response, we develop *MaGNET*, a novel and theoretically motivated latent space sampler for any pre-trained DGN that produces samples uniformly distributed on the learned manifold. We perform a range of experiments on several datasets and DGNs, e.g., for the state-of-the-art StyleGAN2 trained on the FFHQ dataset, uniform sampling via MaGNET increases distribution precision by 4.1% and recall by 3.0% and decreases gender bias by 41.2%, without requiring labels or retraining. Since uniform sample distribution does not imply uniform *semantic* distribution, we also explore how semantic attributes of generated samples vary under MaGNET sampling. Colab and codes at `bit.ly/magnet-sampling`

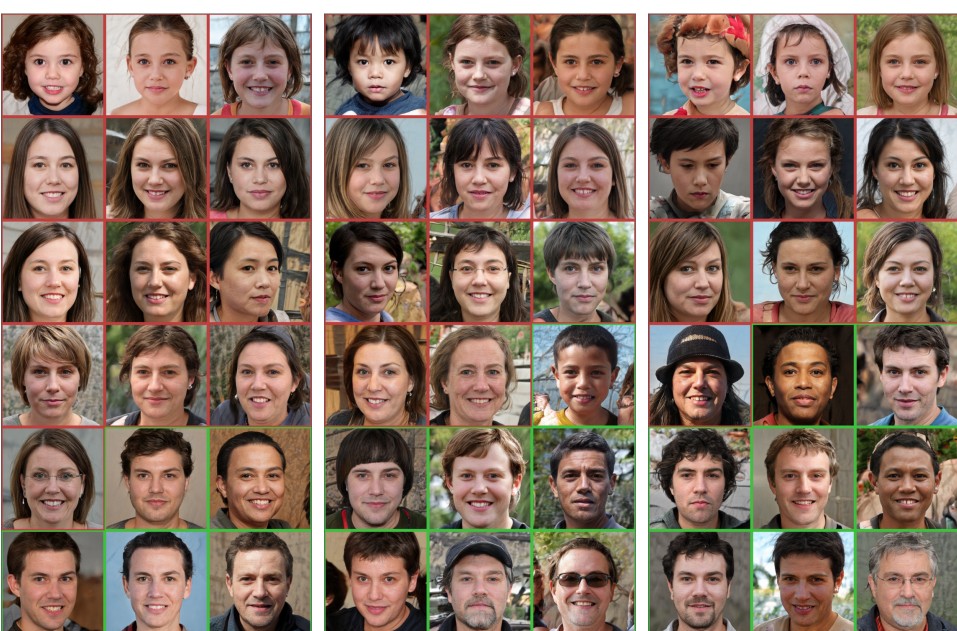

Figure 1: Random batches of StyleGAN2 ($\psi = 0.5$) samples with $1024 \times 1024$ resolution, generated using standard sampling (**left**), uniform sampling via MaGNET on the learned pixel-space manifold (**middle**), and uniform sampling on the style-space manifold (**right**) of the same model. MaGNET sampling yields a higher number of young faces, better gender balance, and greater background/accessory variation, without the need for labels or retraining. Images are sorted by gender-age and color coded red-green (female-male) according to Microsoft Cognitive API predictions. Larger batches of images and attribute distributions are furnished in Appendix E.

# 1 INTRODUCTION

Deep Generative Networks (DGNs) are Deep Networks (DNs) trained to learn latent representations of datasets; such frameworks include Generative Adversarial Networks (GANs) (Goodfellow et al., 2014), Variational Autoencoders (VAEs) (Kingma & Welling, 2013), flow-based models such as NICE (Dinh et al., 2014), and their variants (Dziugaite et al., 2015; Zhao et al., 2016; Durugkar et al., 2017; Arjovsky et al., 2017; Mao et al., 2017; Yang et al., 2019; Fabius & van Amersfoort, 2014; van den Oord et al., 2017; Higgins et al., 2017; Tomczak & Welling, 2017; Davidson et al., 2018; Dinh et al., 2017; Grathwohl et al., 2018; Kingma & Dhariwal, 2018). A common assumption that we will carry through our study is that the datasets of interest are not uniformly distributed in their ambient space, but rather are concentrated on, or around, manifolds of lower intrinsic dimension, e.g., the manifold of natural images (Peyré, 2009). Different DGN training methods have been developed and refined to obtain models that approximate as closely as possible the training set distribution. This becomes an Achilles heel when the training set, regardless of its size, is not representative of the true data distribution, i.e., when the training samples have been curated based on cost or availability that result in implicit/explicit biases. In such scenarios, while the training samples will lie on the true data manifold, the density distribution of the training set will be different from the natural distribution of the data.

*Deploying a DGN trained with a biased data distribution can be catastrophic*, in particular, when employed for tasks such as data augmentation (Sandfort et al., 2019), controlled data generation for exploration/interpretation (Thirumuruganathan et al., 2020), or estimation of statistical quantities of the data geometry, such as the Lipschitz constant of the data manifold (Gulrajani et al., 2017; Scaman & Virmaux, 2018). Biased data generation from DGNs due to skewed training distributions also raises serious concerns in terms of fair machine learning (Hwang et al., 2020; Tan et al., 2020).

While ensuring *semantic uniformity* in samples is an extremely challenging task, we take one step in the more reachable goal of controlling the DGN sampling distribution to be uniform *in terms of the sample distribution on the data manifold*. To that end, *we propose* **MaGNET** *(for **Ma**ximum entropy **G**enerative **NET**work), a simple and efficient modification to any DGN that adapts its latent space distribution to provably produce samples uniformly distributed on the learned DGN manifold*. Importantly, *MaGNET can be employed on any pre-trained and differentiable DGN regardless of its training setting*, reducing the requirement of fine-tuning or retraining of the DGN. This is crucial as many models, such as BigGAN (Brock et al., 2019) and StyleGAN (Karras et al., 2020), have significant computational and energy requirements for training. A plug-and-play method is thus greatly preferred to ease deployment in any already built/trained deep learning pipeline.

Previously, there has been rigorous work on DGNs aimed at improving the training stability of models, deriving theoretical approximation results, understanding the role of the DGN architectures, and numerical approximations to speed-up training and deployment of trained models (Mao et al., 2017; Chen et al., 2018; Arjovsky & Bottou; Miyato et al., 2018; Xu & Durrett, 2018; Liu et al., 2017; Zhang et al., 2017; Biau et al., 2018; Li et al., 2017; Kodali et al., 2017; Roy et al., 2018; Andrés-Terré & Lió, 2019; Chen et al., 2018; Balestriero et al., 2020; Tomczak & Welling, 2016; Berg et al., 2018). Existing methods (Metz et al., 2016; Tanaka, 2019; Che et al., 2020) also try to tackle mode dropping by improving approximation of the data distribution, but this can potentially increase the bias learned implicitly by the DGN. *We are the first to consider the task of providing uniform sampling on the DGN underlying manifold,* which has far-reaching consequences, ranging from producing DGNs stable to data curation and capable of handling inconsistencies such as repeated samples in the training set. We provide a first-of-its-kind provable uniform sampling on the data manifold that can be used to speed up estimation of various geometric quantities, such as estimation of the Lipschitz constant.

MaGNET applies to any (pretrained) DGN architecture (GAN, VAE, NF, etc.) using continuous piecewise affine (CPA) nonlinearities, such as the (leaky) ReLU; smooth nonlinearities can be dealt with via a first-order Taylor approximation argument. Our main contributions are as follows:
**[C1]** We characterize the transformation incurred by a density distribution when composed with a CPA mapping (Sec. 3.1) and derive the analytical sampling strategy that enables one to obtain a uniform distribution on a manifold that is continuous and piecewise affine (Sec 3.2).
**[C2]** We observe that current DGNs produce CPA manifolds, and we demonstrate how to leverage [C1] to produce uniform sampling on the manifold of any DGN (Sec. 3.2).
**[C3]** We conduct several carefully controlled experiments that validate the importance of uniform

sampling and showcase the performance of MaGNET on pretrained models such as BigGAN (Brock et al., 2019), StyleGAN2 (Karras et al., 2020), progGAN (Karras et al., 2017), and NVAE (Vahdat & Kautz, 2020), e.g., we show that MaGNET can be used to increase distribution precision by 4% and recall by 3% for StyleGAN2 and decrease gender bias by 41%, without requiring labels or retraining (Sec. 4.2 and Sec. 4.3).

Plug and play codes for various models are made available at our `Github` repository. Computation and software details are provided in Appendix H, with the proofs of our results in Appendix I. Discussion of the settings in which MaGNET is desirable and possible limitations is provided in Sec. 5.

## 2 BACKGROUND

**Continuous Piecewise Affine (CPA) Mappings.** A rich class of functions emerges from piecewise polynomials: spline operators. In short, given a partition $\Omega$ of a domain $\mathbb{R}^S$, a spline of order $k$ is a mapping defined by a polynomial of order $k$ on each region $\omega \in \Omega$ with continuity constraints on the entire domain for the derivatives of order $0, \ldots, k-1$. As we will focus on affine splines ($k=1$), we only define this case for concreteness. An affine spline $\boldsymbol{S}$ produces its output via

$$\boldsymbol{S}(\boldsymbol{z}) = \sum_{\omega \in \Omega} (\boldsymbol{A}_\omega \boldsymbol{z} + \boldsymbol{b}_\omega) \mathbb{1}_{\{\boldsymbol{z} \in \omega\}}, \tag{1}$$

with input $\boldsymbol{z}$ and $\boldsymbol{A}_\omega, \boldsymbol{b}_\omega$ the per-region *slope* and *offset* parameters respectively, with the key constraint that the entire mapping is continuous over the domain $\boldsymbol{S} \in \mathcal{C}^0(\mathbb{R}^S)$. Spline operators and especially affine spline operators have been extensively used in function approximation theory (Cheney & Light, 2009), optimal control (Egerstedt & Martin, 2009), statistics (Fantuzzi et al., 2002), and related fields.

**Deep Generative Networks.** A deep generative network (DGN) is a (nonlinear) operator $\boldsymbol{G}_\Theta$ with parameters $\Theta$ mapping a *latent* input $\boldsymbol{z} \in \mathbb{R}^S$ to an *observation* $\boldsymbol{x} \in \mathbb{R}^D$ by composing $L$ intermediate *layer* mappings. The only assumption we require for our study is that the nonlinearities present in the DGN are CPA, as is the case with (leaky-)ReLU, absolute value, max-pooling. For smooth nonlinearities, our results hold from a first-order Taylor approximation argument. Precise definitions of DGN operators can be found in Goodfellow et al. (2016). We will omit $\Theta$ from the $\boldsymbol{G}_\Theta$ operator for conciseness unless needed. It is also common to refer to $\boldsymbol{z}$ as the *latent representation*, and $\boldsymbol{x}$ as the *generated/observed data*, e.g., a time-series or image. One property of DGNs that employ nonlinearities such as (leaky-)ReLU, max-pooling, and the likes, is that the entire input-output mapping becomes a CPA spline.

## 3 CONTINUOUS PIECEWISE AFFINE MAPPING OF A PROBABILITY DENSITY

In this section, we study the properties of a probability density that is transformed by a CPA mapping. Our goal is to derive the produced density and characterize its properties, such as how the per-region affine mappings in Eq. 1 impact the density concentration. We present some key results that serve as the backbone of our core result in the next section: how to sample uniformly from the manifold generated by DGNs.

### 3.1 DENSITY ON THE GENERATED MANIFOLD

Consider an affine spline operator $\boldsymbol{S}$ (Eq. 1) going from a space of dimension $S$ to a space of dimension $D$ with $D \geq S$. The image of this mapping is a CPA manifold of dimension at most $S$, the exact dimension is determined by the rank of the per-region slope matrices. Formally, the span, or the image, of $\boldsymbol{S}$ is given by

$$Im(\boldsymbol{S}) \triangleq \{\boldsymbol{S}(\boldsymbol{z}) : \boldsymbol{z} \in \mathbb{R}^S\} = \bigcup_{\omega \in \Omega} \text{Aff}(\omega; \boldsymbol{A}_\omega, \boldsymbol{b}_\omega) \tag{2}$$

with $\text{Aff}(\omega; \boldsymbol{A}_\omega, \boldsymbol{b}_\omega) = \{\boldsymbol{A}_\omega \boldsymbol{z} + \boldsymbol{b}_\omega : \boldsymbol{z} \in \omega\}$ the affine transformation of region $\omega$ by the per-region parameters $\boldsymbol{A}_\omega, \boldsymbol{b}_\omega$.

From Eq. 2 ,we observe that the generated manifold surface is made of regions that are the affine transformations of the latent space partition regions $\omega \in \Omega$ based on the coordinate change induced by $\boldsymbol{A}_\omega$ and the shift induced by $\boldsymbol{b}_\omega$. We visualize this in Fig. 2 for a toy spline operator with a

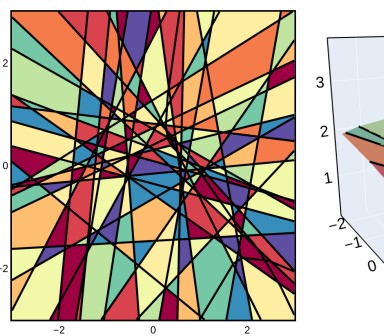 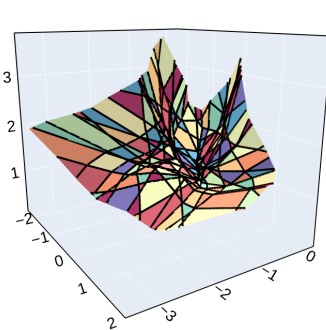

Figure 2: Visual depiction of Eq. 2 with a toy affine spline mapping $\boldsymbol{S}$ : $\mathbb{R}^2 \mapsto \mathbb{R}^3$. **Left:** latent space partition $\Omega$ made of different regions shown with different colors and with boundaries shown in black. **Right:** affine spline image $Im(\boldsymbol{S})$ which is a continuous piecewise affine surface composed of the latent space regions affinely transformed by the per-region affine mappings (Eq. 1). The per-region colors maintain correspondence from the left to the right.

2-dimensional latent space and 3-dimensional ambient/output space. In the remainder of our study we will denote for conciseness $\boldsymbol{S}(\omega) \triangleq \text{Aff}(\omega; \boldsymbol{A}_\omega, \boldsymbol{b}_\omega)$.

When the input space is equipped with a density distribution, then this density is transformed by the mapping $\boldsymbol{S}$ and "lives" on the surface of the CPA manifold generated by $\boldsymbol{S}$. Given a distribution $\boldsymbol{p_z}$ over the latent space, we can explicitly compute the output distribution after the application of $\boldsymbol{S}$, which leads to an intuitive result exploiting the CPA property of the generator. For this result, we require that the operator $\boldsymbol{S}$ be bijective between its domain and range. That is, each slope matrix $\boldsymbol{A}_\omega, \forall \omega \in \Omega$ should be full rank, and there should not be any folding of the generated CPA surface that intersects with itself, i.e., $\boldsymbol{S}(\omega) \cap \boldsymbol{S}(\omega') \neq \{\} \iff \omega = \omega'$. We now derive the key result of this section that characterizes the density distribution on the manifold.

**Lemma 1.** *The volume of a region $\omega \in \Omega$ denoted by $\mu(\omega)$ is related to the volume of the affinely transformed region $\boldsymbol{S}(\omega)$ by*

$$\frac{\mu(\boldsymbol{S}(\omega))}{\mu(\omega)} = \sqrt{\det(\boldsymbol{A}_\omega^T \boldsymbol{A}_\omega)}, \tag{3}$$

*where $\mu(\boldsymbol{S}(\omega))$ is the measure on the S-dimensional affine subspace spanned by the CPA mapping. (Proof in Appendix I.1.)*

**Theorem 1.** *The probability density $p_{\boldsymbol{S}}(\boldsymbol{x})$ generated by $\boldsymbol{S}$ for latent space distribution $\boldsymbol{p_z}$ is given by,*

$$p_{\boldsymbol{S}}(\boldsymbol{x}) = \sum_{\omega \in \Omega} \frac{\boldsymbol{p_z}\left(\left(\boldsymbol{A}_\omega^T \boldsymbol{A}_\omega\right)^{-1} \boldsymbol{A}_\omega^T \left(\boldsymbol{x} - \boldsymbol{b}_\omega\right)\right)}{\sqrt{\det(\boldsymbol{A}_\omega^T \boldsymbol{A}_\omega)}} \mathbb{1}_{\{\boldsymbol{x} \in \boldsymbol{S}(\omega)\}}. \tag{4}$$

*(Proof in Appendix I.2.)*

In words, the distribution obtained in the output space naturally corresponds to a piecewise affine transformation of the original latent space distribution, weighted by the change in volume of the per-region mappings from Eq. 3. For Gaussian and Uniform distributed $\boldsymbol{p_z}$, we use the above results to obtain the analytical form of the density covering the output manifold, we have provided proof and differential entropy derivations in Appendix B.

### 3.2 Making the Density on the Manifold Uniform

The goal of this section is to build on Thm. 1 to provide a novel latent space distribution such that the density distribution lying on the generated manifold is uniform.

One important point that we highlight is that having a Uniform density distribution in the latent space of the affine spline is not sufficient to have a uniform density lying on the manifold; it would be if $\det(\boldsymbol{A}_\omega^T \boldsymbol{A}_\omega) = \det(\boldsymbol{A}_{\omega'}^T \boldsymbol{A}_{\omega'}), \forall \omega \neq \omega'$ (in words, the change in volume of the per region mapping is equal for all $\omega$). This is evident from Appendix B (Eq. 8). Therefore we propose here a novel latent space sampler with the purpose that once it is transformed by the affine spline (i.e., the DGN) a distribution becomes uniform on the DGN manifold. We focus here on the technical aspect and defer precise motivations behind such construction to the next section that deals with practical applications. To obtain $K$ samples uniformly distributed on the output manifold of $\boldsymbol{S}$ using the proposed MaGNET procedure:

1. For $K$ MaGNET samples, sample $N \gg K$ (as large as possible) iid latent vectors with $U$ being the latent space domain of $\boldsymbol{S}$ ($\boldsymbol{z}_1, \ldots, \boldsymbol{z}_N$), with $\boldsymbol{z}_i \sim \mathcal{U}(U)$.

2. Compute the per-region slope matrices $\boldsymbol{A}_i \triangleq \boldsymbol{J_S}(\boldsymbol{z}_i)$ (Eq. 1), and the change of volume scalar $(\sigma_1, \ldots, \sigma_N) \triangleq \left( \sqrt{\det(\boldsymbol{A}_1^T \boldsymbol{A}_1)}, \ldots, \sqrt{\det(\boldsymbol{A}_N^T \boldsymbol{A}_N)} \right)$, where $\boldsymbol{A}_i = \boldsymbol{A}_\omega \mathbb{1}_{\{z_i \in \omega\}}$ .

3. Sample (with replacement) $K$ latent vectors $(\boldsymbol{z}_1, \ldots, \boldsymbol{z}_K)$ with probability $\propto (\sigma_1, \ldots, \sigma_N)$

We discuss possible choices of $N$ and $K$ in Appendix D, where we observe that even for state-of-the-art models like StyleGAN2, $N =$250,000 is sufficient to provide a stable approximation of the true latent space target distribution. In practice, $\boldsymbol{A}_i$ is simply obtained through backpropagation, since it is the Jacobian matrix of the DGN at $\boldsymbol{z}_i$, as in $\boldsymbol{A}_i = \boldsymbol{JS}(\boldsymbol{z}_i)$.

The above Monte-Carlo approximation does not require knowledge of the DGN spline partition $\Omega$ nor the per-region slope matrices (Eq. 1). Those are computed on-demand as $\boldsymbol{z}_i$ are sampled. The above procedure produces uniform samples on the manifold learned by a DGN regardless of how it has been trained.

## 4 MaGNET: Maximum Entropy Generative Network Sampling

The goal of this section is to first bridge current DGNs with affine splines & leverage Thm. 1 and Sec. 3.2 to effectively produce uniform samples on the manifold of DGNs such as BigGAN, Style-GAN. We build this affine spline DGN bridge and motivate for uniform sampling in Sec. 4.1 and present various experiments across architectures in Sec. 4.2, 4.3, and 4.4.

### 4.1 Uniform Sampling on the Deep Generative Network Manifold

We provided in Sec. 3.2 a thorough study of affine splines and how those mappings transform a given input distribution. This now takes high relevance as per the following remark.

**Remark 1.** *Any DGN (or part of it) that employs CPA nonlinearities (as in Sec. 2) is itself a CPA; that is, the input-output mapping can be expressed as in (Eq. 1).*

This observation in the context of classifier DNs goes back to Montufar et al. (2014) and has been further studied in Unser (2018); Balestriero & Baraniuk (2018). We also shall emphasize that operators such as Batch-Normalization (Ioffe & Szegedy, 2015) are not continuous piecewise affine during training but become affine operators during evaluation time. For completeness, we also provide that analytical form of the per-region affine mappings $\boldsymbol{A}_\omega, \boldsymbol{b}_\omega$ of Eq. 1 for the DGNs featured Appendix C. The key for our method is thus to combine the above with the results from Sec. 3.2 to obtain the following statement.

**Theorem 2.** *Consider a training set sampled from a manifold $\mathcal{M}$ and a (trained) CPA DGN $\boldsymbol{S}$. As long as $\mathcal{M} \subset Im(\boldsymbol{S})$, sampling from $\boldsymbol{S}$ as per Sec. 3.2 produces uniform samples on $\mathcal{M}$, regardless of the training set sampling. (Proof in I.4.)*

This result follows by leveraging the analytical DGN distribution from Thm. 1 and by replacing $\boldsymbol{p_z}$ with the proposed one, leading to $\boldsymbol{p}_S(\boldsymbol{x}) \propto \sum_{\omega \in \Omega} \mathbb{1}_{\{\boldsymbol{x} \in \boldsymbol{S}(\omega)\}}$ which is uniform on the DGN manifold. By using the above *one can take any (trained) DGN and produce uniform samples on the learned underlying manifold*. Hence, our solution produces a generative process that becomes invariant to the training set distribution. While this provides a theoretical guarantee for uniform sampling, it also highlights the main limitation of MaGNET: *the uniform samples will lie on a CPA manifold*. That is, unless the true manifold $\mathcal{M}$ is also continuous, MaGNET will occasionally introduce abnormal samples that correspond to sampling from the regions of discontinuity of $\mathcal{M}$. We will see in the following sections how even on high-quality image datasets, MaGNET produces very few abnormal samples, one reason being that for complicated data manifolds, state-of-the-art DGNs are often built with (class) conditioning. In such cases, the above continuity assumption on $\mathcal{M}$ lessens only to a within-class continuity assumption which is much more realistic. Sampling uniformly on the DGN manifold has many important applications that are deferred to the following sections.

### 4.2 Quantitative Validation: $\epsilon$-Ball Concentration, GMM Likelihood and Fréchet Inception Distance

We now report three controlled experiments to validate the applicability of the theoretical results from Sec. 3.2 for the MaGNET sampling procedure.

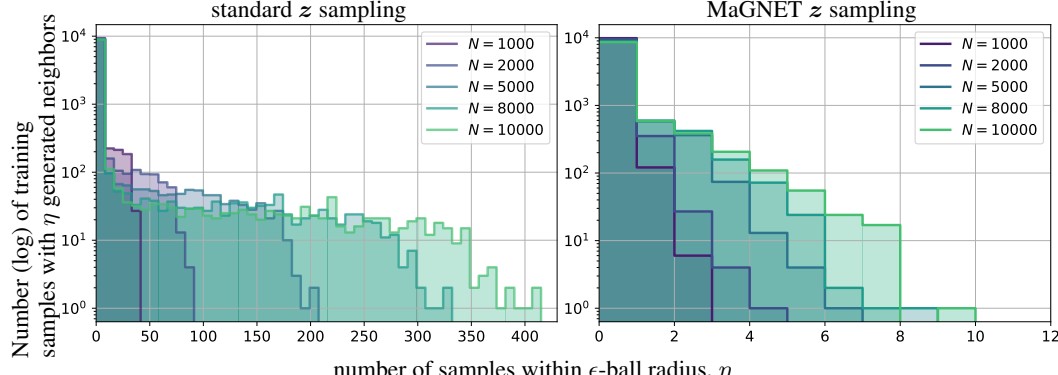

Figure 3: Distribution of the number of MNIST training samples with $\eta$ neighbors generated within an $\epsilon$-ball radius. $N$ samples are generated using standard sampling and MaGNET sampling using an NVAE model. Here $\epsilon$ is taken to be the average nearest neighbor distance for the training samples. For Vanilla NVAE, heavier tails are indicative of larger density variations on the manifold as $N$ is increased, whereas for MaGNET the shorter tails are indicative of fewer variations in neighborhood density, i.e., uniform generation on the data manifold. 10,000 MNIST samples are used for comparison, for additional $\epsilon$ see Fig. 13 and Fig. 14 in the Appendix.

First, we consider MNIST and assume that the entire data manifold is approximately covered by the training samples. Regardless of the training data distribution on the manifold (uniform or not), we can pick a datum at random, count how many generated samples ($\eta$) are within this datum $\epsilon$-ball neighborhood and repeat this process for 10,000 training samples. If $\eta$ does not vary between training datum, then it strongly indicates that the generated samples are uniformly distributed on the manifold covered by the training data. We perform this experiment using a pretrained state-of-the-art variational autoencoder NVAE (Vahdat & Kautz, 2020) to compare between standard and MaGNET sampling with the number of generated samples $N$ ranging from 1,000 to 10,000. We report the distribution of $\eta$ in Fig. 3. Again, uniform sampling is equivalent to having the same $\eta$ for all training samples, i.e., a Dirac distribution in the reported histograms. We can see that MaGNET sampling approaches that distribution while standard sampling has a heavy-tail $\eta$ distribution, i.e., the generated digits have different concentrations at different parts of the data manifold. Another quantitative measure consists of fitting a Gaussian Mixture Model (GMM) with varying number of clusters, on the generated data, and comparing the likelihood obtained for standard and MaGNET sampling. As we know that in both cases the samples lie on the same manifold and domain, the sampling with lower likelihood will correspond to the one for which samples are spread more uniformly on the manifold. We report this in Fig. 4, further confirming the ability of MaGNET to produce uniformly spread samples. We report the generated samples in Appendix E. Lastly, we compare the Fréchet Inception Distance (FID) (Heusel et al., 2017) between 50,000 generated samples and 70,000 training samples for StyleGAN2 (config-f) trained on FFHQ. Since uniform sampling via MaGNET increases the diversity of generated samples, we see that MaGNET sampling improves the FID for truncation (Karras et al., 2019), $\psi = \{.4, .5, .6, .7\}$ by 2.76 points on average (see Appendix F). While for the aforementioned $\psi$ MaGNET samples alone provide an improved FID, for higher $\psi$ values, we introduce an increasing amount of MaGNET samples for FID calculation. We observe in Fig. 4 that by progressively increasing the percentage of MaGNET samples, we are able to exceed the state-of-the-art FID of 2.74 for StyleGAN2 ($\psi = 1$), reaching an FID of 2.66 with $\sim 4\%$ of MaGNET samples.

### 4.3 Qualitative Validation: High-Dimensional State-of-the-art Image Generation

We now turn into the qualitative evaluation of MaGNET sampling, to do so we propose extensive experiments on various state-of-the-art image DGNs. We also remind the reader that in all cases, standard and MaGNET sampling are performed on the same DGN (same weights) as discussed in Sec. 3.2.

**2-Dimensional Dataset and Colored-MNIST.** The first set of controlled experiments is designed such that the training set contains inconsistencies while it is known that the original distribution is uniform on the data manifold. Such inconsistencies can occur in real datasets due to challenges related to dataset compilation. We provide illustrative examples in Fig. 5, where we demonstrate

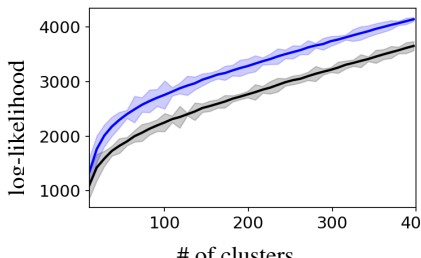 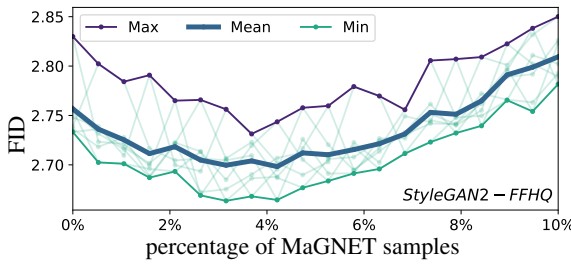

Figure 4: (Left) Average log-likelihood and $10\sigma$-bandwidth for 5 runs of a GMM trained on 10,000 samples using standard sampling (blue) and MaGNET sampling (black) for an NVAE trained on MNIST. The higher log-likelihood (given the same number of clusters) in the standard sampling case demonstrates an increased concentration around a few modes, as opposed MaGNET. (Right) FID ($\downarrow$) of StyleGAN2 (config-f) trained on FFHQ for 50,000 generated samples and 7 runs. With an increasing percentage of uniformly generated samples to increase diversity, MaGNET reaches state-of-the-art FID of 2.66 achieved at a 4.1% mixture.

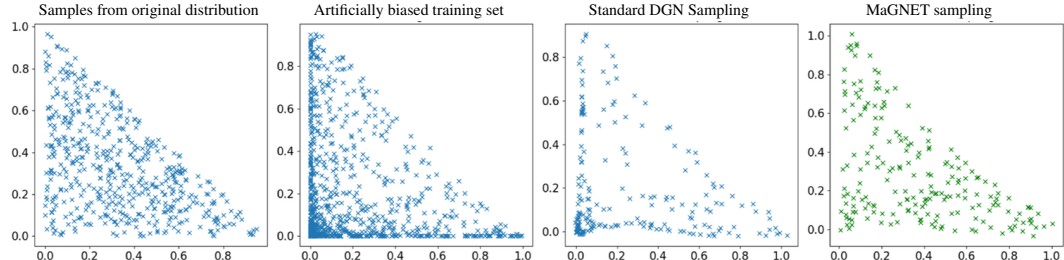

Figure 5: *From left to right*, samples from a toy 2D distribution with triangular support, biased samples obtained for training a GAN, standard sampling showing a biased distribution learned by the GAN, and MaGNET sampling recovering uniformly distributed samples on the support of the true distribution. Note that the same number of samples are obtained for both standard and MaGNET sampling.

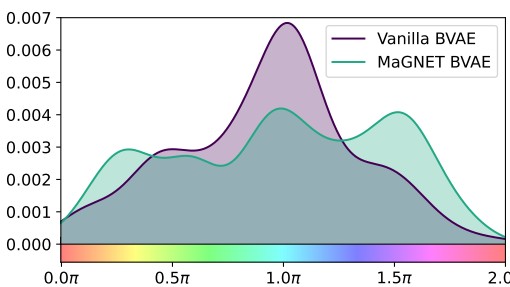

Figure 6: Hue (color) distribution of samples obtained from standard and MaGNET sampling from a trained BVAE model on colored 8 digits from MNIST. The original dataset purposefully favored cyan coloring to represent training set inconsistency. MaGNET BVAE can approximately generate uniformly colored MNIST samples. The density drop around red represent regions where training density was low, therefore the DGN manifold approximation was incorrect and uniform sampling can not recover those samples.

that unless uniform sampling is employed, the trained DGN reproduces the inconsistencies present in the training set, as expected. This toy dataset visualization validates our method from Sec. 3.2.

Going further, we take the MNIST dataset (in this case, only digit 8 samples) and apply imbalanced coloring based on the hue distribution provided in Appendix Fig. 12, which favors cyan color. We train a $\beta$-VAE DGN (BVAE) on that cyan-inclined dataset, and present in Fig. 6 the hue distributions for samples obtained via standard sampling and MaGNET sampling. We observe that MaGNET corrects the hue distribution back to uniformity.

**Uniform Face Generation: CelebA-HQ and Flickr-Faces-HQ with progGAN and Style-GAN2.** Our first experiment concerns sampling from the StyleGAN2 (Karras et al., 2020) model pretrained on the Flickr-Faces-HQ (FFHQ) dataset. StyleGAN2 has two DGNs, one that maps to an intermediate latent space, termed style-space and another DGN that maps style-space vectors to the pixels-space (output of StyleGAN2). Implementation details are contained in Appendix H. We focus here on applying MaGNET onto the entire StyleGAN2 model (the composition of both DGNs), in Sec. 4.4 we discuss applying MaGNET to the style-space DGN. In Fig. 1 we provide random samples from the same StyleGAN2 model obtained via standard and MaGNET sampling. Upon qualitative evaluation, it can be seen that the samples obtained via MaGNET (MaGNET Style-GAN2) have a significantly larger variety of age distribution, background variations and wearable

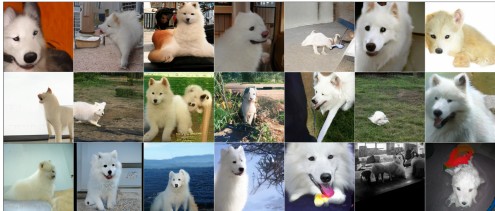
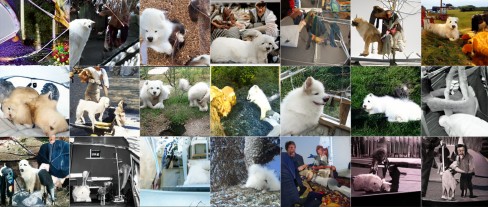

Figure 7: Random batch of samples generated from BigGAN (left) and MaGNET BigGAN (right), conditioned on the *Samoyed* class of ImageNet. While BigGAN samples contain homogeneous postures, MaGNET samples represent the true span of the data manifold learned by BigGAN.

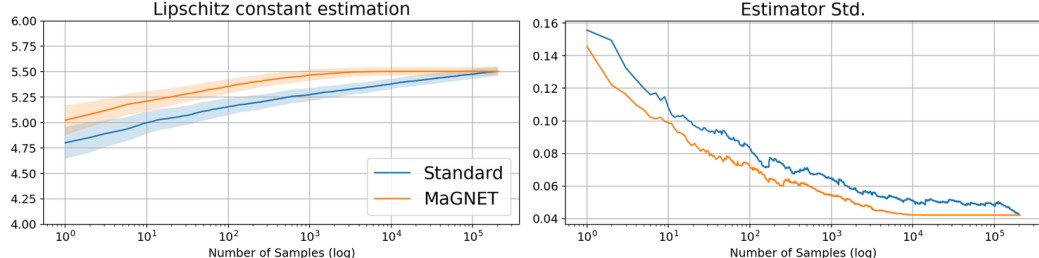

Figure 8: Lipschitz constant estimation using standard DGN sampling and MaGNET sampling (left). Each line represents the mean over 200 Monte-Carlo runs. As expected, estimation of such statistics from samples converges faster when employed on uniformly distributed samples. The standard deviation of the Monte-Carlo estimations are also provided (right), where it is clear that the uniform sampling via MaGNET reaches smaller standard deviation at earlier steps i.e., it is faster to converge. This is a key application of MaGNET: speeding-up convergence of statistical estimation of quantities, such as the Lipschitz constant in this case. The x-axis represents the number of samples used for estimation in log-scale.

accessories compared to standard sampling.

For experiments with the CelebA-HQ dataset, we adopt the Progressively Growing GAN (prog-GAN) (Karras et al., 2017), trained on $1024 \times 1024$ resolution images. In Fig. 9 we provide random samples from standard and MaGNET sampling, the latter portraying more qualitative diversity. We see that uniform manifold sampling via MaGNET recovers samples containing a number of attributes that are generally underrepresented in the samples generated by vanilla progGAN. (See Appendix E for larger batches and attribute distributions.) Note that uniform sampling not only recovers under-represented groups e.g., age $< 30$, head-wear, and bald hair, it also increases the presence of neutral emotion and black hair. One interesting observation is that MaGNET also increases the number of samples off the true data manifold (images that are not celebrity faces), exposing regions where the manifold is not well approximated by progGAN.

**Conditionally Uniform Generation: ImageNet with BigGAN.** We present experiments on the state-of-the-art conditional generative model BigGAN (Brock et al., 2019) using MaGNET sampling. In Fig. 7 we provide random samples from standard and MaGNET sampling. More experiments on different classes are presented in Appendix E. We see that uniform sampling on the learned data manifold yields a large span of backgrounds and textures, including humans, while standard sampling produces examples closer to the modes of the training dataset. This is quite understandable considering that ImageNet was curated using a large number of images scraped from the internet. MaGNET therefore could be used for data exploration/model interpretation and also as a diagnostic tool to assess the quality of the learned manifold a posteriori of training.

### 4.4 APPLICATION: MONTE-CARLO ESTIMATION AND ATTRIBUTE REBALANCING

We conclude this section with two more practical aspects of MaGNET.

**Reduced-Variance Monte-Carlo Estimator.** The first is to speed-up (in terms of number of required samples) basic Monte-Carlo estimation of arbitrary topological quantities of the generated manifold. Suppose that one's goal is to estimate the Lipschitz constant of a DGN. A direct estimation method would use the known bound given by $\max_{\boldsymbol{z}} \|\boldsymbol{J_S}(\boldsymbol{z})\|_F$ (Wood & Zhang, 1996). This estimation can be done by repeatedly sampling latent vectors $\boldsymbol{z}$ from the same distribution that one used for training a DGN. However, this implies that the produced samples will not be uniformly distributed on the manifold in turn leading to slower convergence of the estimator. Instead, we propose to use MaGNET, and report our findings in Fig. 8. More domains of application, where MaGNET

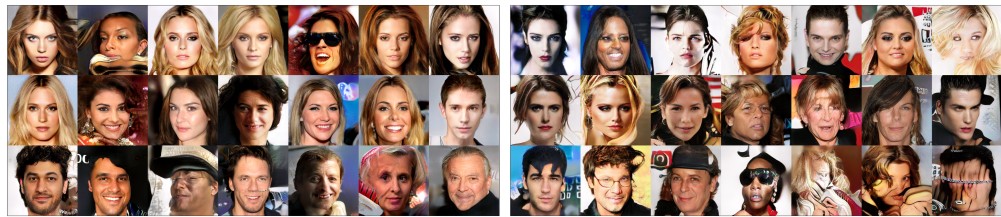

Figure 9: Random batch of samples generated from vanilla progGAN (left) and MaGNET progGAN (right). Some samples contain fusion of attributes which are not frequently represented in the data distribution, e.g. female w/ facial hair, male w/ makeup, etc. MaGNET also generates more samples from regions of the manifold less approximated by the DGN. Larger batches in Appendix E.

can be used for estimator variance reduction, can be found in Baggenstoss (2017).

**Style-space MaGNET sampling rebalances attributes.** When thinking of uniform sampling on a manifold, it might seem natural to expect fairness i.e., fair representation of different attributes such as equal representation of gender, ethnicity, hair color, etc. However, this is not necessarily true in all cases. In fact, it is trivial to show that each attribute category will be equally represented iff their support on the true data manifold is of equal volume (integrated with respect to the data manifold). Fortunately, as we mentioned in Sec. 4.3, architectures such as StyleGAN2 have explicitly built a style-space, which is a latent space in which attributes are organized along affine subspaces occupying similar volumes (Karras et al., 2019) i.e., MaGNET applied on the style-space DGN should improve fairness. By applying MaGNET sampling on the style-space, we are able to reduce gender bias from 67–33% (female-male) in standard StyleGAN2 to 60–40%. This simple result demonstrates the importance of our proposed sampling and how it can be used to increase fairness for DGNs trained on biased training sets. MaGNET in the style-space also yields improvements in terms of recall and precision (Sajjadi et al., 2018). Given a reference distribution (e.g., FFHQ dataset) and a learned distribution, precision measures the fidelity of generated samples while recall measures diversity. We compare the metrics for face images generated via $z \sim N(0, aI)$ where $a \in 0.5, 1, 1.5, 2, z \sim U[-2, 2]$, and MaGNET sampling on style-space. For 70k samples generated for each case, MaGNET sampling obtains a recall and precision of $(0.822, 0.92)$ with a 4.12% relative increase in recall and 3.01% relative increase in precision compared to the other latent sampling methods (metrics were averaged for 10 seeds).

## 5 Conclusions, Limitations and Future Work

We have demonstrated how the affine spline formulation of DGN provides new theoretical results to provably provide uniform sampling on the manifold learned by a DGN. This allows becoming robust to possibly incorrect training set distributions that any DGN would learn to replicate after its training. We have reported on several experiments using pretrained state-of-the-art generative models and demonstrated that uniform sampling on the manifold offers many benefits from data exploration to statistical estimation. Beyond the sole goal of uniform sampling on a manifold, MaGNET opens many avenues, yet MaGNET is not a "one size fits all" solution.

**When not to sample uniformly.** We can identify the general cases in which one should not employ uniform sampling of the DGN manifold. The first case occurs whenever the true manifold is known to be discontinuous and one needs to avoid sampling in those regions of discontinuities. In fact, in the discontinuous case, DGN training will adapt to put zero (or near zero) density in those discontinuous regions preventing standard sampling to reach those regions (Balestriero et al., 2020). However, MaGNET will reverse this process and introduce samples back in those regions. The second case occurs if one aims to produce samples from the same distribution as the training set distribution (assuming training of the DGN was successful). In this scenario, one should use the same latent distribution at evaluation time as the one used during training.

**Future work.** Currently, there are two main limitations of our MaGNET sampling strategy. The first one lies in the assumption that the trained DGN is able to learn a good enough approximation of the true underlying data manifold. In future work, we plan to explore how MaGNET can be used to test such an assumption. One potential direction is as follows; train a DGN using several sub-sampled datasets (similar to bootstrap methods) and then study if MaGNET samples populate manifolds that all coincide between the different DGNs. If training is successful, then those sampled manifolds should coincide. Another direction could be understanding the relationship between uniform sampling and uniform attribute representation. We demonstrated how uniform sampling in the style-space of StyleGAN2 ensures that relationship by construction.

## 6 REPRODUCIBILITY STATEMENT

Reproducible data and code for various experiments is made available at bit.ly/magnet-sampling. Computation and software details are provided in Appendix H, with the proofs of our results in Appendix I.

## ACKNOWLEDGEMENTS

This work was supported by NSF grants CCF-1911094, IIS-1838177, and IIS-1730574; ONR grants N00014-18-12571, N00014-20-1-2534, and MURI N00014-20-1-2787; AFOSR grant FA9550-22-1-0060; and a Vannevar Bush Faculty Fellowship, ONR grant N00014-18-1-2047.

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

# SUPPLEMENTARY MATERIAL

The following appendices support the main paper and are organized as follows: Appendix A addresses some details on the background of continuous piecewise affine deep networks, that were omitted in the main paper. Appendices E and H provide additional figures and training details for all the experiments that were studied in the main text, and Appendix I provides the proofs of all the theoretical results. **Due to filesize constraints, the high-quality batch of samples are provided in the supplementary files**.

## A  BACKGROUND ON CONTINUOUS PIECEWISE AFFINE DEEP NETWORKS

A *max-affine spline operator* (MASO) concatenates independent *max-affine spline* (MAS) functions, with each MAS formed from the point-wise maximum of $R$ affine mappings (Magnani & Boyd, 2009; Hannah & Dunson, 2013). For our purpose each MASO will express a DN layer and is thus an operator producing a $D^\ell$ dimensional vector from a $D^{\ell-1}$ dimensional vector and is formally given by

$$\text{MASO}(\boldsymbol{v}; \{\boldsymbol{A}_r, \boldsymbol{b}_r\}_{r=1}^R) = \max_{r=1,\dots,R} \boldsymbol{A}_r \boldsymbol{v} + \boldsymbol{b}_r, \tag{5}$$

where $\boldsymbol{A}_r \in \mathbb{R}^{D^\ell \times D^{\ell-1}}$ are the slopes and $\boldsymbol{b}_r \in \mathbb{R}^{D^\ell}$ are the offset/bias parameters and the maximum is taken coordinate-wise. For example, a layer comprising a fully connected operator with weights $\boldsymbol{W}^\ell$ and biases $\boldsymbol{b}^\ell$ followed by a ReLU activation operator corresponds to a (single) MASO with $R=2, \boldsymbol{A}_1 = \boldsymbol{W}^\ell, \boldsymbol{A}_2 = \boldsymbol{0}, \boldsymbol{b}_1 = \boldsymbol{b}^\ell, \boldsymbol{b}_2 = \boldsymbol{0}$. Note that a MASO is a *continuous piecewise-affine* (CPA) operator (Wang & Sun, 2005).

The key background result for this paper is that *the layers of DNs constructed from piecewise affine operators (e.g., convolution, ReLU, and max-pooling) are MASOs* (Balestriero & Baraniuk, 2018):

$$\exists R \in \mathbb{N}^*, \exists\{\boldsymbol{A}_r, \boldsymbol{b}_r\}_{r=1}^R \text{ s.t. } \text{MASO}(\boldsymbol{v}; \{\boldsymbol{A}_r, \boldsymbol{b}_r\}_{r=1}^R) = g^\ell(\boldsymbol{v}), \forall \boldsymbol{v} \in \mathbb{R}^{D^{\ell-1}}, \tag{6}$$

making the entire DGN a composition of MASOs.

The CPA spline interpretation enabled from a MASO formulation of DGNs provides a powerful global geometric interpretation of the network mapping based on a partition of its input space $\mathbb{R}^S$ into polyhedral regions and a per-region affine transformation producing the network output. The partition regions are built up over the layers via a *subdivision* process and are closely related to Voronoi and power diagrams (Balestriero et al., 2019). We now propose to greatly extend such insights to carefully characterize and understand DGNs as well as provide theoretical justifications to various observed behaviors e.g. mode collapse.

## B  UNIFORM AND GAUSSIAN MANIFOLD DISTRIBUTIONS

We now demonstrate the use of the above result by considering practical examples for which we are able to gain insights into the DGN data modeling and generation. We consider the two most common cases: (i) the latent distribution is set as $\boldsymbol{z} \sim \mathcal{N}(0,1)$ and (ii) the latent distribution is set as $\boldsymbol{z} \sim \mathcal{U}(0,1)$ (on the hypercube of dimension $S$). We obtain the following result by direct application of Thm. 1.

**Corollary 1.** *The generated density distribution $p_{\boldsymbol{S}}$ of the Gaussian and uniform densities are given by*

$$p_{\boldsymbol{S}}(\boldsymbol{x}) = \sum_{\omega \in \Omega} \frac{e^{-\frac{1}{2}(\boldsymbol{x}-\boldsymbol{b}_\omega)^T (\boldsymbol{A}_\omega^+)^T \boldsymbol{A}_\omega^+ (\boldsymbol{x}-\boldsymbol{b}_\omega)}}{\sqrt{(2\pi)^S \det(\boldsymbol{A}_\omega^T \boldsymbol{A}_\omega)}} \mathbb{1}_{\{\boldsymbol{x} \in \boldsymbol{G}(\omega)\}}, \qquad \textit{(Gaussian)} \tag{7}$$

$$p_{\boldsymbol{S}}(\boldsymbol{x}) = \sum_{\omega \in \Omega} \frac{Vol(U)^{-1}}{\sqrt{\det(\boldsymbol{A}_\omega^T \boldsymbol{A}_\omega)}} \mathbb{1}_{\{\boldsymbol{x} \in \boldsymbol{S}(\omega)\}}. \qquad \textit{(Uniform)} \tag{8}$$

The two above formulae provide a precise description of the produced density given that the latent space density is Gaussian or Uniform. In the Gaussian case, the per-region slope matrices act upon

the $\ell_2$ distance by rescaling it from the coordinates of $\boldsymbol{A}_\omega$ and the per-region offset parameters $\boldsymbol{b}_\omega$ are the mean against which the input $\boldsymbol{x}$ is compared against. In the Uniform case, the change of volume (recall Eq. 3) is the only quantity that impacts the produced density. We will heavily rely on this observation for the next section where we study how to produce a uniform sampling onto the CPA manifold of an affine spline.

We derive the analytical form for the case of Gaussian and Uniform latent distribution in Appendix I.3. From the analytical derivation of the generator density distribution, we obtain its differential entropy.

**Corollary 2.** *The differential Shannon entropy of the output distribution $\boldsymbol{p_G}$ of the DGN is given by* $E(\boldsymbol{p_G}) = E(\boldsymbol{p_z}) + \sum_{\omega \in \Omega} P(\boldsymbol{z} \in \omega) \log(\sqrt{\det(\boldsymbol{A}_\omega^T \boldsymbol{A}_\omega)})$.

As a result, the differential entropy of the output distribution $\boldsymbol{p_G}$ corresponds to the differential entropy of the latent distribution $\boldsymbol{p_z}$ plus a convex combination of the per-region volume changes. It is thus possible to optimize the latent distribution $\boldsymbol{p_z}$ to better fit the target distribution entropy as in Ben-Yosef & Weinshall (2018) and whenever the prior distribution is fixed, any gap between the latent and output distribution entropy imply the need for high change in volumes between $\omega$ and $\boldsymbol{G}(\omega)$.

## C   PER-REGION AFFINE MAPPINGS

For completeness we also provide that analytical form of the per-region affine mappings

$$\boldsymbol{A}_\omega = \left( \prod_{i=0}^{L-1} \text{diag}\left( \dot{\sigma}^{L-i}(\omega) \right) \boldsymbol{W}^{L-i} \right), \tag{9}$$

$$\boldsymbol{b}_\omega = \boldsymbol{b}^L + \sum_{\ell=1}^{L-1} \left[ \left( \prod_{i=0}^{L-\ell-1} \text{diag}\left( \dot{\sigma}^{L-i}(\omega) \right) \boldsymbol{W}^{L-i} \right) \text{diag}\left( \dot{\sigma}^{\ell}(\omega) \right) \boldsymbol{b}^\ell \right], \tag{10}$$

where $\dot{\sigma}^\ell(\boldsymbol{z})$ is the pointwise derivative of the activation function of layer $\ell$ based on its input $\boldsymbol{W}^\ell \boldsymbol{z}^{\ell-1} + \boldsymbol{b}^\ell$, which we note as a function of $\boldsymbol{z}$ directly. For precise definitions of those operators see Balestriero & Baraniuk (2020). The diag operator simply puts the given vector into a diagonal square matrix. For convolutional layers (or else) one can simply replace the corresponding $\boldsymbol{W}^\ell$ with the correct slope matrix parametrization as discussed in Sec. 2. Notice that since the employed activation functions $\sigma^\ell, \forall \ell \in \{1, \ldots, L\}$ are piecewise affine, their derivative is piecewise constant, in particular with values $[\dot{\sigma}^\ell(\boldsymbol{z})]_k \in \{\alpha, 1\}$ with $\alpha = 0$ for ReLU, $\alpha = -1$ for absolute value, and in general with $\alpha > 0$ for Leaky-ReLU for $k \in \{1, \ldots, D^\ell\}$.

## D   NUMBER OF SAMPLES AND UNIFORMITY

Exact uniformity is reached when the Monte Carlo samples have covered each region of the DGN partition boundary. For large state-of-the-art models this condition requires sampling on the order of millions. However, we conducted an experiment to see how the number of samples really impacted the uniformity of the generated manifold as follows. We compute precision and recall metrics [4] for StyleGAN2 with $K$ generated samples obtained from $N$ Monte Carlo samples based on our sampling strategy by varying $N$. We use $K = 5000$ and $N$ ranging from 10,000 to 500,000. Based on the metrics, we identify that increasing beyond $K = 250,000$ no longer impacts the metrics, showing that this number of monte carlo samples is enough to converge (approximately) to the uniform sampling in that case; see Fig. 10.

We report here the Jacobian computation times for Tensorflow 2.5 with CUDA 11 and Cudnn 8 on an NVIDIA Titan RTX GPU. For StyleGAN2 pixel space, 5.03s/it; StyleGAN2 style-space, 1.12s/it; BigGAN 5.95s/it; ProgGAN 3.02s/it. For NVAE on Torch 1.6 it takes 20.3s/it. Singular value calculation for StyleGAN2 pixel space takes 0.005s/it, StyleGAN2 style space 0.008s/it, BigGAN 0.001s/it, ProgGAN 0.004s/it and NVAE 0.02s/it on NumPy.

## E   ADDITIONAL FIGURES

This section contains samples from our proposed methods, more samples along with attribute data and pretrained weights are available at our project `link`.

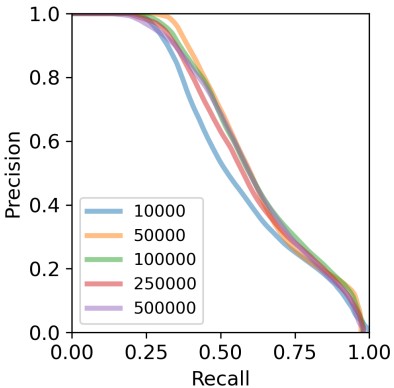

Figure 10: Evolution of the precision/recall curves for varying number of samples $N$ form the monte-carlo sampling against the number of samples $K = 5k$ for StyleGAN2.

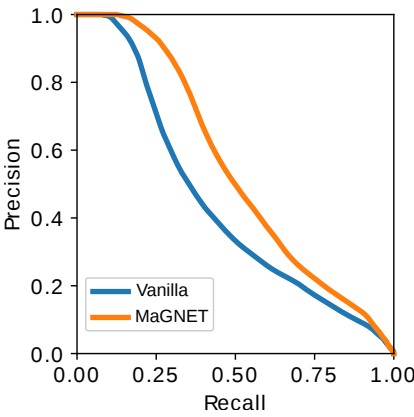

Figure 11: Precision-recall curves for $K = 70k$ samples from Vanilla StyleGAN2 and MaGNET StyleGAN2

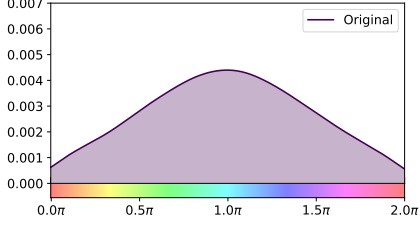

Figure 12: Depiction of the imbalance hue distribution applied to color the MNIST digits.

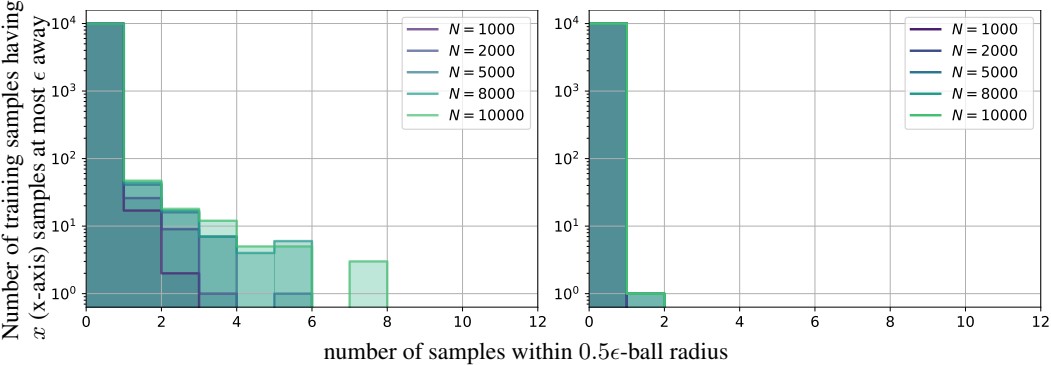

Figure 13: Reprise of Fig. 3. Vanilla NVAE **Left**, MaGNET NVAE **Right**

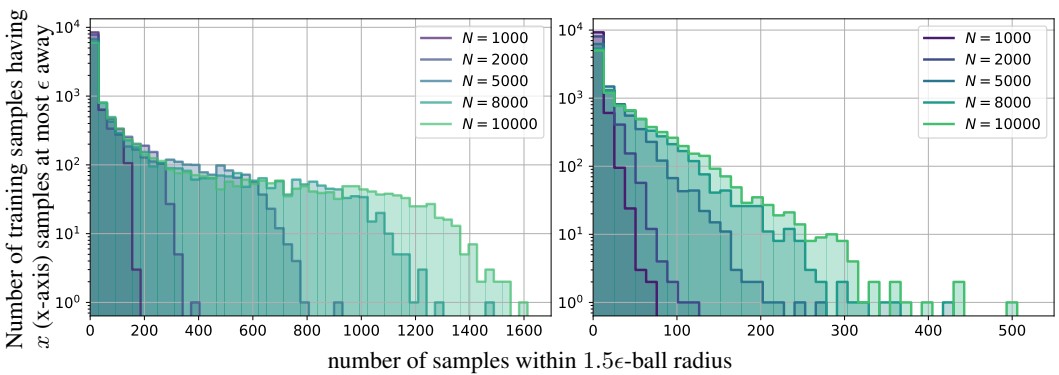

Figure 14: Reprise of Fig. 3. Vanilla NVAE **Left**, MaGNET NVAE **Right**

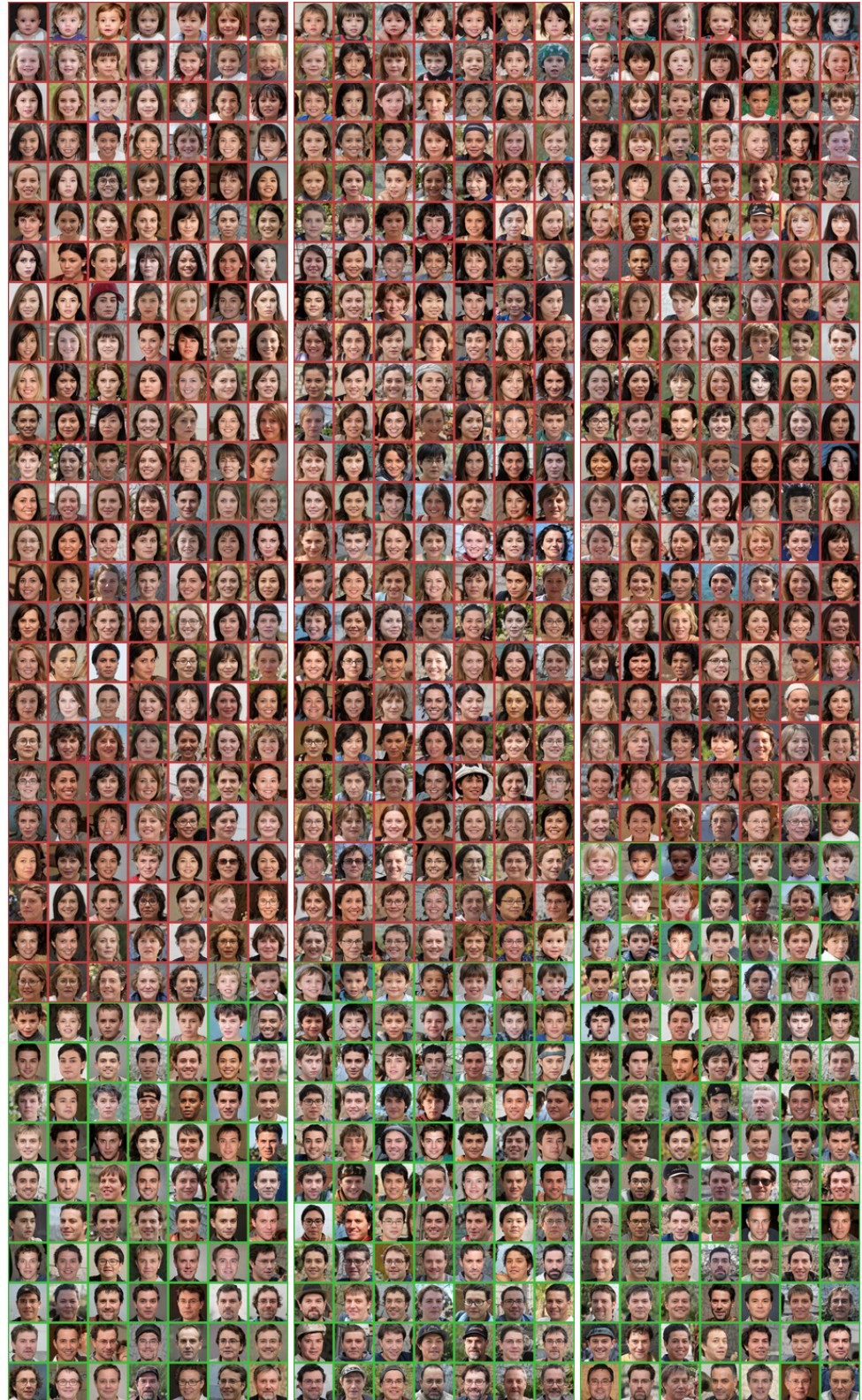

Figure 15: Random batches of 245 samples from a StyleGAN2 trained on FFHQ, generated via standard sampling (left), MaGNET sampling in the pixel-space (middle) and MaGNET sampling in the style-space.

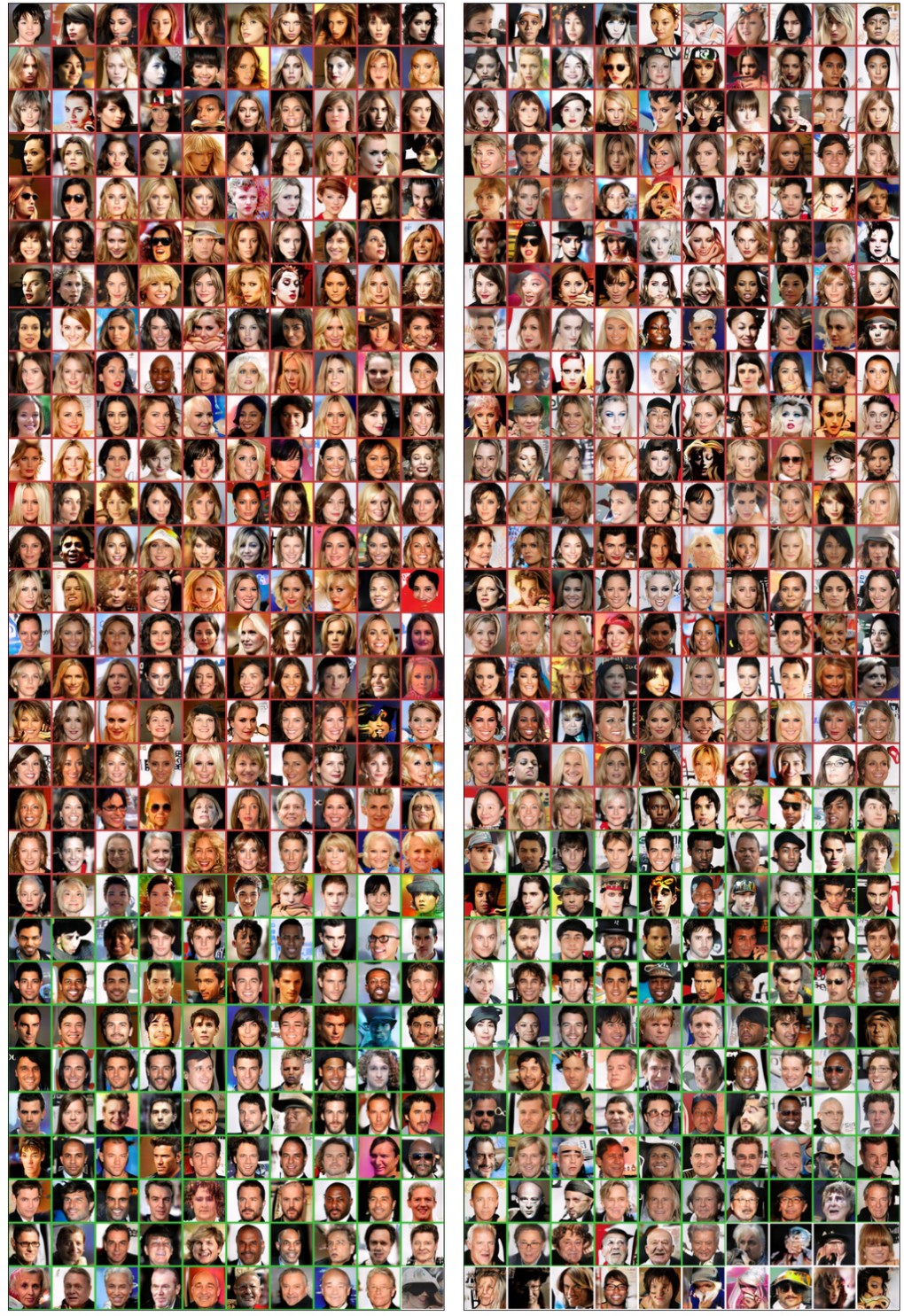

Figure 16: Random Samples from vanilla progGAN (left) and MaGNET progGAN (right) trained on the CelebA-HQ dataset. Samples are sorted by gender & age and color coded by gender as visually predicted by the Microsoft Cognitive API. Samples not recognized by the API are color coded as white at the bottom.

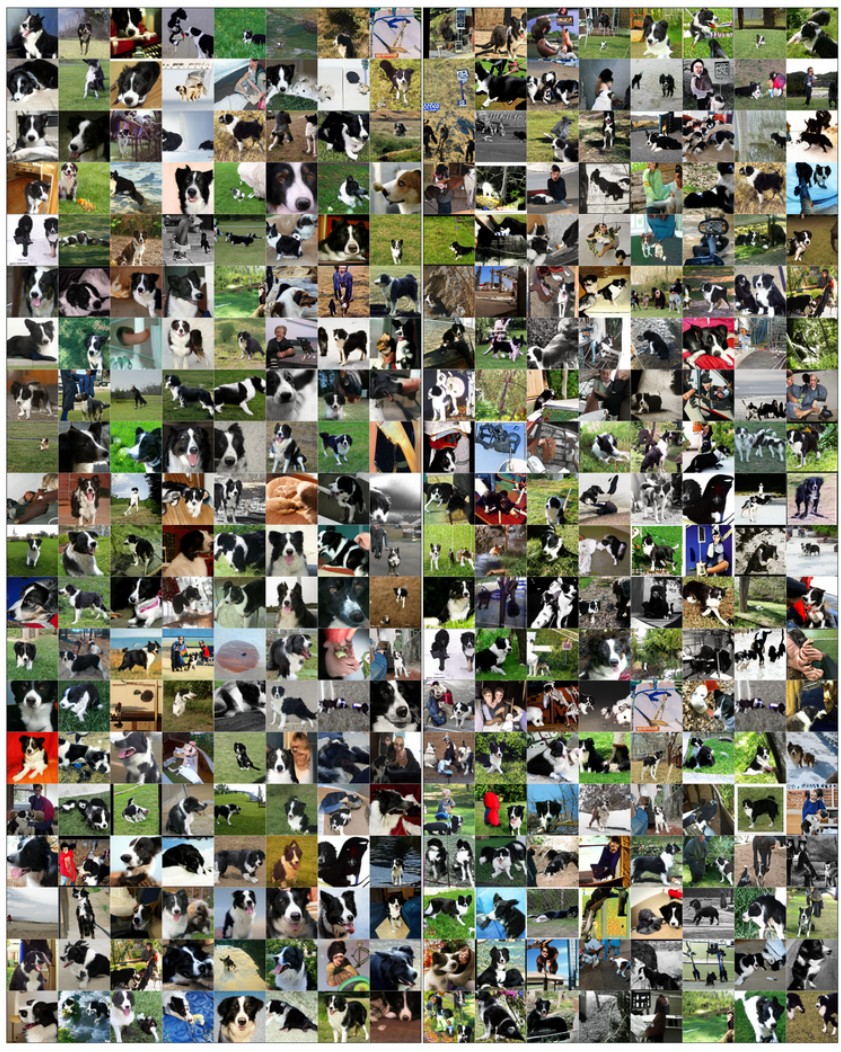

Figure 17: Random Samples from vanilla BigGAN (left) and MaGNET BigGAN (right) from the *Collie* class of Imagenet.

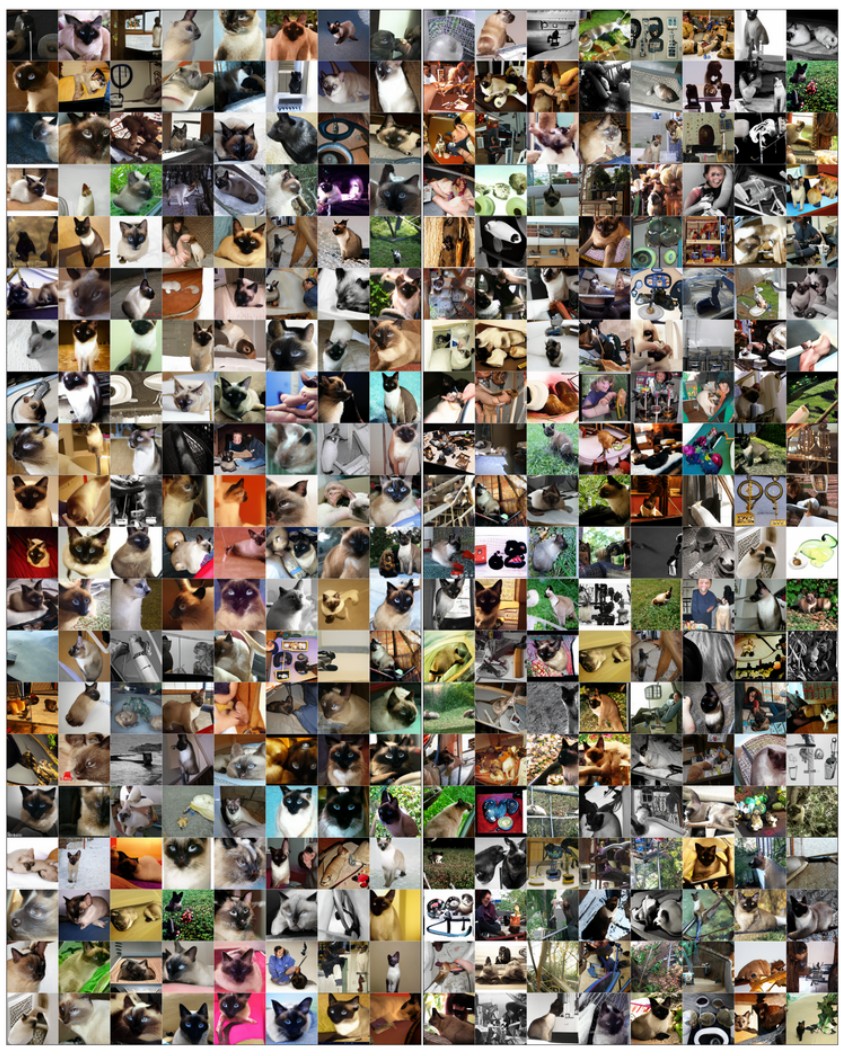

Figure 18: Random Samples from vanilla BigGAN (left) and MaGNET BigGAN (right) from the *Siamese* class of Imagenet.

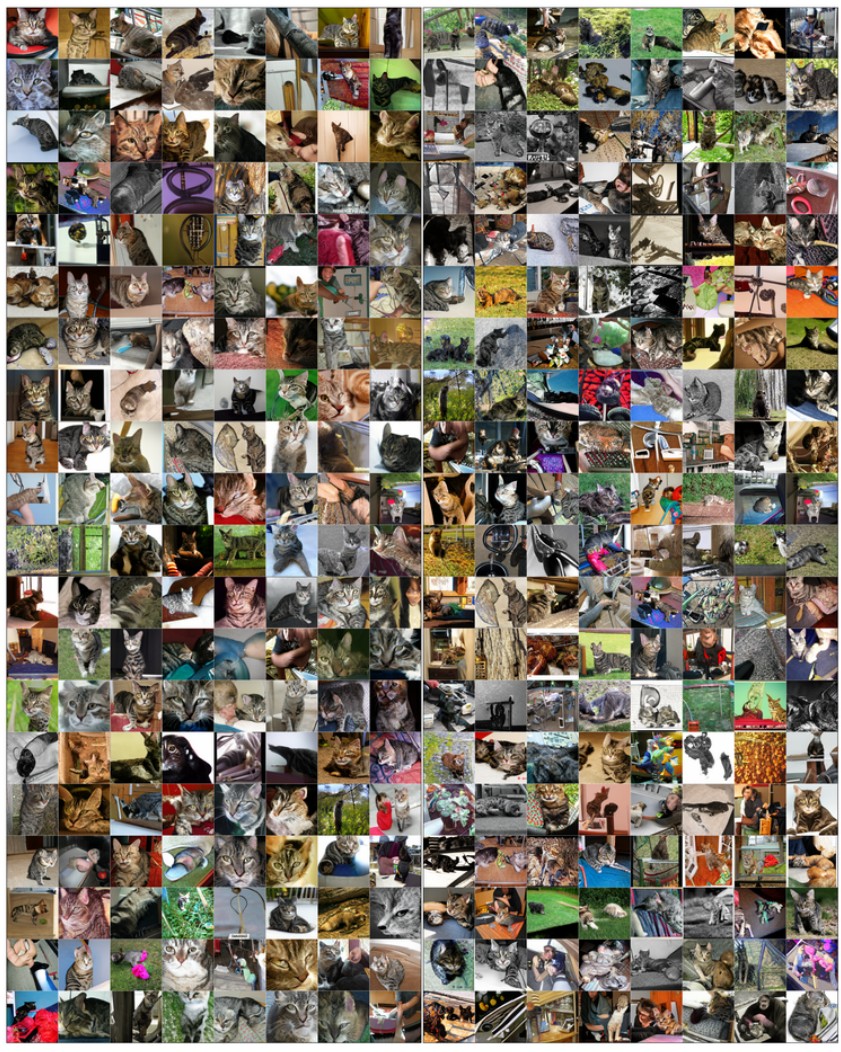

Figure 19: Random Samples from vanilla BigGAN (left) and MaGNET BigGAN (right) from the *Tabby* class of Imagenet.

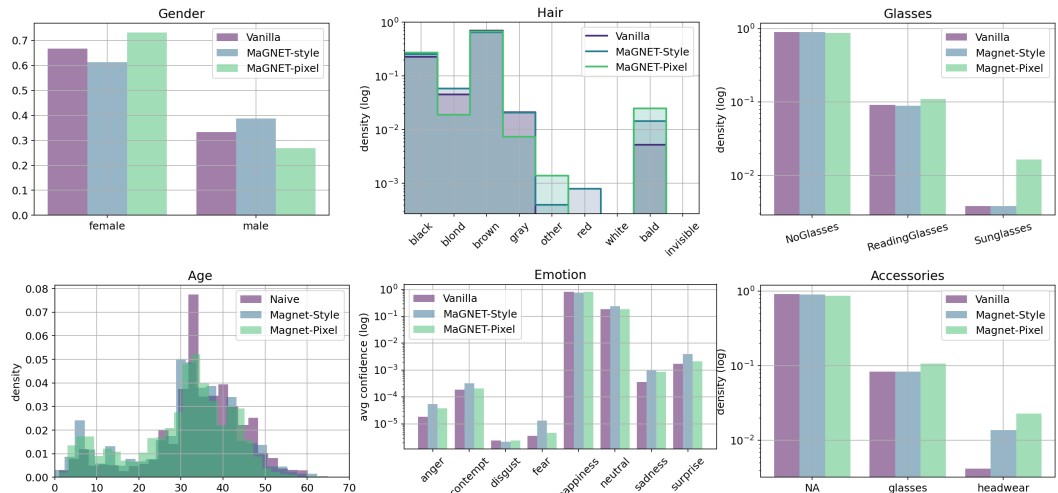

Figure 20: Facial Attributes of 5000 StyleGAN2 samples using vanilla sampling, MaGNET style-space sampling and MaGNET pixel-space sampling. We see that MaGNET style-space increases uniformity in gender and age distributions whereas MaGNET pixel-space yields more variations in physical attributes and accessories.

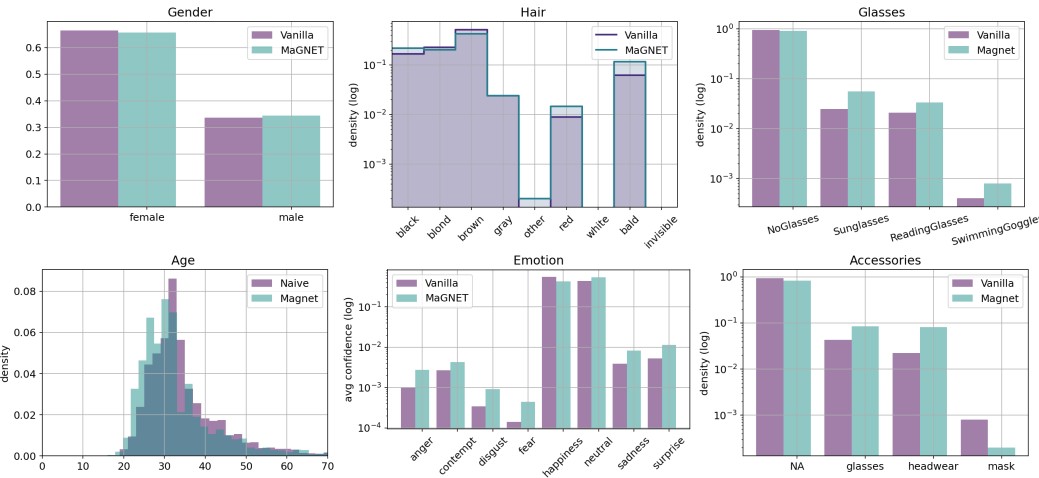

Figure 21: Facial Attributes of 5000 ProgGAN samples using standard sampling and MaGNET sampling.

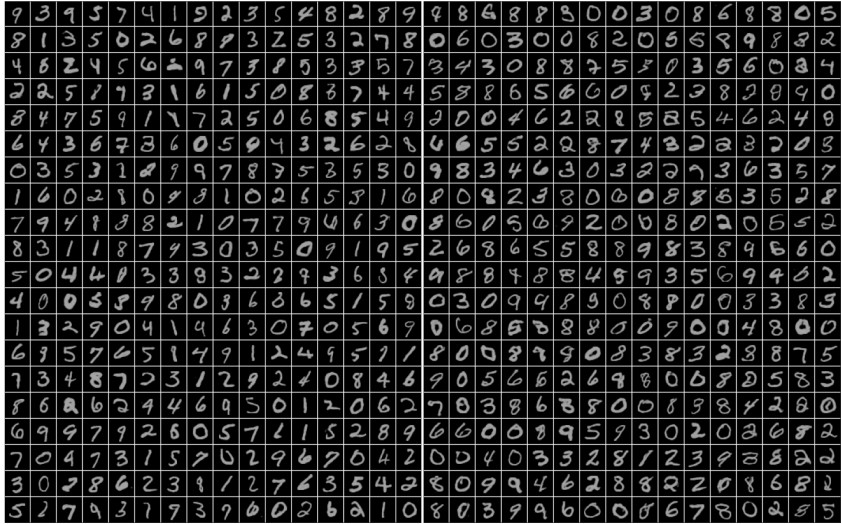

Figure 22: Random Samples from vanilla NVAE (left) and MaGNET NVAE (right) trained on the MNIST dataset.

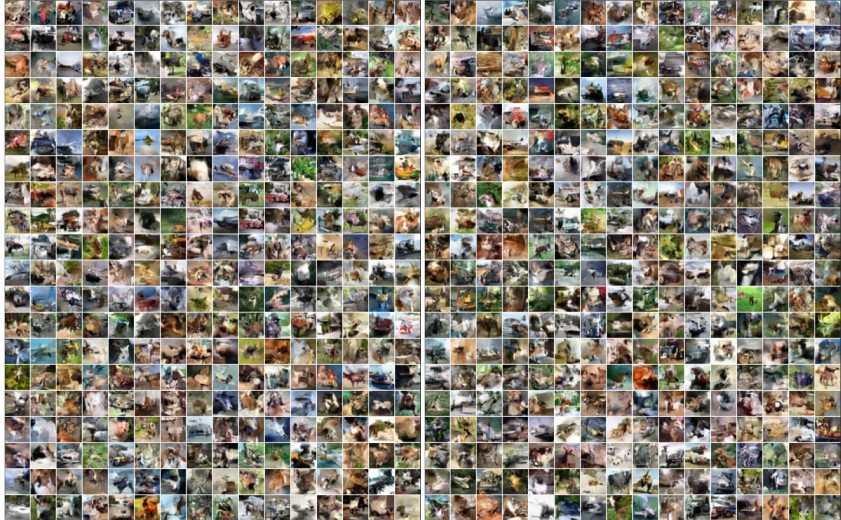

Figure 23: Random Samples from vanilla NVAE (left) and MaGNET NVAE (right) trained on the CIFAR dataset.

# F  ADDITIONAL TABLES

Table 1: FID obtained by StyleGAN2 (config-f) trained on FFHQ using standard and MaGNET sampling (pixel-space) for varying degrees of truncation ($\psi$). (Left) FID score obtained for 50,000 samples generated by a mixture of MaGNET and standard sampling. MaGNET samples can be used to increase diversity of the model, resulting in better FID than current state-of-the-art. (Right) FID score for varying truncation without mixing.

| Truncation $\psi$ | Sampling Method | Percent Mixture | FID ($\downarrow$) | Truncation $\psi$ | Sampling Method | FID ($\downarrow$) |
|---|---|---|---|---|---|---|
| 1 | Standard | - | 2.74 | .5 | Standard | 58.33 |
| | MaGNET | 4.1% | **2.66** | | MaGNET | **54.47** |
| .9 | Standard | - | 5.05 | .4 | Standard | 83.84 |
| | MaGNET | 20% | **4.29** | | MaGNET | **82.41** |
| .8 | Standard | - | 10.94 | .3 | Standard | **112.08** |
| | MaGNET | 33% | **8.57** | | MaGNET | 112.89 |
| .7 | Standard | - | 21.34 | .2 | Standard | **142.27** |
| | MaGNET | 100% | **19.41** | | MaGNET | 144.93 |
| .6 | Standard | - | 36.98 | .1 | Standard | **176.20** |
| | MaGNET | 100% | **33.19** | | MaGNET | 178.75 |

# G  ALGORITHMS

---

**Algorithm 1:** MaGNET Sampling as described in Sec. 3.2

---

**Input:** Latent space domain, $U$; Generator $\boldsymbol{G}$; Number of regions to sample $N$; Number of samples $K$;
**Output:** MaGNET Samples, $\{x_i\}_{i=1}^{K}$;
Initialize, $\mathcal{Z}, \mathcal{S} \leftarrow [], []$ ;
**for** $n = 1, \ldots, N$ **do**
     $z \sim \mathcal{U}(U)$;
     Get Slope Matrix, $\boldsymbol{A} = \boldsymbol{J_G}(z)$;
     Get volume scalar at $z$, $\sigma_z = \sqrt{det(\boldsymbol{A}^T\boldsymbol{A})}$;
     $\mathcal{Z}$.append($z$);
     $\mathcal{S}$.append($\sigma_z$)
**end**
**for** $n = 1, \ldots, K$ **do**
     $i \sim \text{Categorical}(\text{prob} = \text{softmax}(\mathcal{S}))$;
     $x_i \leftarrow \mathcal{Z}[i]$
**end**

---

**Algorithm 2:** Online Rejection Sampling algorithm for MaGNET

---

**Input:** Latent space domain, $U$; Generator $\boldsymbol{G}$; $N$ change of volume scalars $\{\sigma_1, \sigma_2, ..., \sigma_N\}$;
**Output:** MaGNET Sample, $x$;
**while** *True* **do**
     Sample $z \sim \mathcal{U}(U)$;
     Sample $\alpha \sim \mathcal{U}[0, 1]$;
     Get Slope Matrix, $\boldsymbol{A} = \boldsymbol{J_G}(z)$;
     Get volume scalar at $z$, $\sigma_z = \sqrt{det(\boldsymbol{A}^T\boldsymbol{A})}$;
     **if** $\frac{\sigma_z}{\sigma_z + \sum_{i=1}^{N} \sigma_i} \geq \alpha$ **then**
         $x = \boldsymbol{G}(z)$;
         **break**;
     **end**
**end**

---

# H    ARCHITECTURE, HARDWARE AND IMPLEMENTATION DETAILS

All the experiments were run on a Quadro RTX 8000 GPU, which has 48 GB of high-speed GDDR6 memory and 576 Tensor cores. For the software details we refer the reader to the provided codebase. In short, we employed TF2 (2.4 at the time of writing), all the usual Python scientific libraries such as NumPy and PyTorch. We employed the official repositories of the various models we employed with official pre-trained weights. As a note, most of the architectures can not be run on GPUs with less or equal to 12 GB of memory.

For StyleGAN2, we use the official config-e provided in the GitHub StyleGAN2 repo[1], unless specified. We use the recommended default of $\psi = 0.5$ as the interpolating stylespace truncation, to ensure generation quality of faces for the qualitative experiments. For BigGAN we use the BigGAN-deep architecture with no truncation, available on TFHub[2]. We also use the NVAE[3] and ProgGAN[4] models and weights from their respective official implementations. For the Jacobian determinant calculation of images w.r.t latents, we first use a random orthogonal matrix to project generated images into a lower dimensional space, calculate the Jacobian of the projection w.r.t the latents and calculate the singular values of the jacobian to estimate the volume scalar. We use a projection of 256 dimensions for StyleGAN2-pixel, ProgGAN and BigGAN, and 128 dimensions for NVAE. To estimate the volume scalar we use the top 30, 20, 15 singular values for StyleGAN2 MaGNET pixel, ProgGAN and BigGAN; 40 for StyleGAN2 MaGNET style, and 30 for NVAE.

# I    PROOFS

## I.1    PROOF OF LEMMA 1

*Proof.*    In the special case of an affine transform of the coordinate given by the matrix $A \in \mathbb{R}^{D \times D}$ the well known result from demonstrates that the change of volume is given by $|\det(A)|$ (see Theorem 7.26 in Rudin (2006)). However in our case the mapping is a rectangular matrix as we span an affine subspace in the ambient space $\mathbb{R}^D$ making $|\det(A)|$ not defined.

First, we shall note that in the case of a Riemannian manifold (as is the produced surface from the per-region affine mapping) the volume form used in the usual change of variable formula can be defined via the square root of the determinant of the metric tensor. Now, for a surface of intrinsic dimension $n$ embedded in Euclidean space of dimension $m$ (in our case, the per-region affine mapping produces an affine subspace) parametrized by the mapping $M : \mathbb{R}^n \mapsto \mathbb{R}^m$ (in our case this mapping is simply the affine mapping $M(z) = z_\omega z + b_\omega$ for each region) the metric tensor is given by $g = DM^T DM$ with $D$ the jacobian/differential operator (in our case $g = A_\omega^T A_\omega$ for each region). This result is also known as Sard's theorem (Spivak, 2018). We thus obtain that the change of volume from the region $\omega$ to the affine subspace $G(\omega)$ is given by $\sqrt{\det(A^T A)}$ which can also be written as follows with $USV^T$ the svd-decomposition of the matrix $A$:

$$
\begin{aligned}
\sqrt{\det(A^T A)} = \sqrt{\det((USV^T)^T(USV^T))} &= \sqrt{\det((VS^T U^T)(USV^T))} \\
&= \sqrt{\det(VS^T SV^T)} \\
&= \sqrt{\det(S^T S)} \\
&= \prod_{i:\sigma_i \neq 0} \sigma_i(A)
\end{aligned}
$$

leading to

$$
\int_{\text{Aff}(\omega, A, b)} dx = \frac{1}{\sqrt{\det(A^T A)}} \int_\omega dz
$$

□

---

[1]https://github.com/NVlabs/stylegan2

[2]https://tfhub.dev/deepmind/biggan-deep-256/1

[3]https://github.com/NVlabs/NVAE

[4]https://github.com/tkarras/progressive_growing_of_gans

## I.2 PROOF OF THEOREM 1

*Proof.* We will be doing the change of variables $\boldsymbol{z} = (\boldsymbol{A}_\omega^T \boldsymbol{A}_\omega)^{-1} \boldsymbol{A}_\omega^T (\boldsymbol{x} - \boldsymbol{b}_\omega) = \boldsymbol{A}_\omega^+ (\boldsymbol{x} - \boldsymbol{b}_\omega)$, also notice that $J_{\boldsymbol{G}^{-1}}(\boldsymbol{x}) = A^+$. First, we know that $P_{\boldsymbol{G}(\boldsymbol{z})}(\boldsymbol{x} \in w) = P_{\boldsymbol{z}}(\boldsymbol{z} \in \boldsymbol{G}^{-1}(w)) = \int_{\boldsymbol{G}^{-1}(w)} p_{\boldsymbol{z}}(\boldsymbol{z})d\boldsymbol{z}$ which is well defined based on our full rank assumptions. We then proceed by

$$P_{\boldsymbol{G}}(\boldsymbol{x} \in w) = \sum_{\omega \in \Omega} \int_{\omega \cap w} p_{\boldsymbol{z}}(\boldsymbol{G}^{-1}(\boldsymbol{x}))\sqrt{\det(J_{\boldsymbol{G}^{-1}}(\boldsymbol{x})^T J_{\boldsymbol{G}^{-1}}(\boldsymbol{x}))}d\boldsymbol{x}$$

$$= \sum_{\omega \in \Omega} \int_{\omega \cap w} p_{\boldsymbol{z}}(\boldsymbol{G}^{-1}(\boldsymbol{x}))\sqrt{\det((\boldsymbol{A}_\omega^+)^T \boldsymbol{A}_\omega^+)}d\boldsymbol{x}$$

$$= \sum_{\omega \in \Omega} \int_{\omega \cap w} p_{\boldsymbol{z}}(\boldsymbol{G}^{-1}(\boldsymbol{x}))(\prod_{i:\sigma_i(\boldsymbol{A}_\omega^+)>0} \sigma_i(\boldsymbol{A}_\omega^+))d\boldsymbol{x}$$

$$= \sum_{\omega \in \Omega} \int_{\omega \cap w} p_{\boldsymbol{z}}(\boldsymbol{G}^{-1}(\boldsymbol{x}))(\prod_{i:\sigma_i(\boldsymbol{A}_\omega)>0} \sigma_i(\boldsymbol{A}_\omega))^{-1}d\boldsymbol{x} \quad \text{Etape 1}$$

$$= \sum_{\omega \in \Omega} \int_{\omega \cap w} p_{\boldsymbol{z}}(\boldsymbol{G}^{-1}(\boldsymbol{x}))\frac{1}{\sqrt{\det(\boldsymbol{A}_\omega^T \boldsymbol{A}_\omega)}}d\boldsymbol{x}$$

Let now prove the Etape 1 step by proving that $\sigma_i(A^+) = (\sigma_i(A))^{-1}$ where we lighten notations as $A := \boldsymbol{A}_\omega$ and $USV^T$ is the svd-decomposition of $A$:

$$A^+ = (A^T A)^{-1}A^T = ((USV^T)^T(USV^T))^{-1}(USV^T)^T$$
$$= (VS^T U^T USV^T)^{-1}(USV^T)^T$$
$$= (VS^T SV^T)^{-1}VS^T U^T$$
$$= V(S^T S)^{-1}S^T U^T$$
$$\implies \sigma_i(A^+) = (\sigma_i(A))^{-1}$$

with the above it is direct to see that $\sqrt{\det((\boldsymbol{A}_\omega^+)^T \boldsymbol{A}_\omega^+)} = \frac{1}{\sqrt{\det(\boldsymbol{A}_\omega^T \boldsymbol{A}_\omega)}}$ as follows

$$\sqrt{\det((\boldsymbol{A}_\omega^+)^T \boldsymbol{A}_\omega^+)} = \prod_{i:\sigma_i \neq 0} \sigma_i(\boldsymbol{A}_\omega^+) = \prod_{i:\sigma_i \neq 0} \sigma_i(\boldsymbol{A}_\omega)^{-1}$$

$$= \left(\prod_{i:\sigma_i \neq 0} \sigma_i(\boldsymbol{A}_\omega)\right)^{-1}$$

$$= \frac{1}{\sqrt{\det(\boldsymbol{A}_\omega^T \boldsymbol{A}_\omega)}}$$

which gives the desired result. $\square$

## I.3 PROOF OF COROLLARY 1

*Proof.* We now demonstrate the use of Thm. 1 where we consider that the latent distribution is set as $\boldsymbol{z} \sim \mathcal{N}(0, 1)$. We obtain that

$$\boldsymbol{p}_{\boldsymbol{G}}(\boldsymbol{x} \in w) = \sum_{\omega \in \Omega} \int_{\omega \cap w} \mathbb{1}_{\boldsymbol{x} \in \boldsymbol{G}(\omega)} p_{\boldsymbol{z}}(\boldsymbol{G}^{-1}(\boldsymbol{x})) \det(\boldsymbol{A}_\omega^T \boldsymbol{A}_\omega)^{-\frac{1}{2}}d\boldsymbol{x}$$

$$= \sum_{\omega \in \Omega} \int_{\omega \cap w} \mathbb{1}_{\boldsymbol{x} \in \boldsymbol{G}(\omega)} \frac{1}{(2\pi)^{S/2}\sqrt{\det(\boldsymbol{A}_\omega^T \boldsymbol{A}_\omega)}} e^{-\frac{1}{2}\|\boldsymbol{G}^{-1}(\boldsymbol{x})\|_2^2}d\boldsymbol{x}$$

$$= \sum_{\omega \in \Omega} \int_{\omega \cap w} \mathbb{1}_{\boldsymbol{x} \in \boldsymbol{G}(\omega)} \frac{1}{(2\pi)^{S/2}\sqrt{\det(\boldsymbol{A}_\omega^T \boldsymbol{A}_\omega)}} e^{-\frac{1}{2}((\boldsymbol{A}_\omega^+(\boldsymbol{x}-\boldsymbol{b}_\omega))^T((\boldsymbol{A}^+(\boldsymbol{x}-\boldsymbol{b}_\omega)))}d\boldsymbol{x}$$

$$= \sum_{\omega \in \Omega} \int_{\omega \cap w} \mathbb{1}_{\boldsymbol{x} \in \boldsymbol{G}(\omega)} \frac{1}{(2\pi)^{S/2}\sqrt{\det(\boldsymbol{A}_\omega^T \boldsymbol{A}_\omega)}} e^{-\frac{1}{2}(\boldsymbol{x}-\boldsymbol{b}_\omega)^T(\boldsymbol{A}_\omega^+)^T \boldsymbol{A}_\omega^+(\boldsymbol{x}-\boldsymbol{b}_\omega)}d\boldsymbol{x}$$

giving the desired result that is reminiscent of Kernel Density Estimation (KDE) (Rosenblatt, 1956) and in particular adaptive KDE (Breiman et al., 1977), where a partitioning of the data manifold is performed on each cell ($\omega$ in our case) different kernel parameters are used. $\quad\square$

*Proof.* We now turn into the uniform latent distribution case on a bounded domain $U$ in the DGN input space. By employing again Thm. 1, the given formula one can directly obtain that the output density is given by

$$p_{\boldsymbol{G}}(\boldsymbol{x}) = \frac{\sum_{\omega\in\Omega}\mathbb{1}_{\boldsymbol{x}\in\omega}\det(\boldsymbol{A}_\omega^T\boldsymbol{A}_\omega)^{-\frac{1}{2}}}{Vol(U)} \tag{11}$$

$\square$

## I.4 PROOF OF THM. 2

*Proof.* As we assume successful training, then regardless of the actual distribution $p_x$, the DGN will learn the correct underlying manifold, and learn the best approximation to $p_x$ as possible onto this manifold. Now, applying MaGNET sampling i.e. Sec. 3.2 is equivalent to sampling from a distribution $p_{\boldsymbol{m}z}$ such that after DGN mapping, that distribution is uniform on the learned manifold (see Thm. 1). As we assumed that regardless of $p_x$ the DGN approximates correctly the true manifold, and as we then adapt the sampling distribution $p_{\boldsymbol{m}z}$ to always obtain uniform sampling on that manifold, we see that this final sampling becomes invariant upon the data distribution (on the manifold) leading to the desired result. $\quad\square$

