# OpenReview forum: "MaGNET: Uniform Sampling from Deep Generative Network Manifolds Without Retraining"
_ICLR.cc/2022/Conference — ICLR 2022 Poster_

### Official Review · Reviewer_5nSv · 2021-10-31

**Correctness:** 3
**Technical Novelty And Significance:** 2
**Empirical Novelty And Significance:** 2
**Recommendation:** 6
**Confidence:** 4

**Main Review:**

This paper proposes a sampling method that aims at providing uniform samples from the latent space for deep generative models. Providing uniform samples from the latent space is very important, but the manuscript requires more quantitative results to support their claims of uniform samples:

Major concerns:
1. How does the MaGNET sampling affect the quality of the generated images? Common GANs literature provides some quantitative evaluation metrics (inception score, FID score, KID score) as a justification for their proposed methods, but this submission does not provide any justification using the prevalent evaluation metrics. The visual quality of some generated images using the MaGNET method is poorer than using the original GAN alone (Figure 7 &9) In addition, in addition, the provided samples of the generated images (Figure 15 - 20) are not clear enough to provide a good justification for their visual quality.

2. It is hard to tell whether the proposed method truly improves the uniformity of the sampled data (in Figure 22):
 (1) (Gender) it seems that the MaGNET-style reduces gender bias but the MaGNET-pixel increases gender bias
 (2) (Hair) due to the low clarity of this subfigure, it is hard to draw any conclusion
 (3) (Glasses) it seems that the MaGNET-pixel improves the occurrence of sunglasses and the improvement of MaGNET-style is limited
 (4) (Age) it seems that both methods have some improvement in age
 (5) (Emotion) the improvement of this subfigure is hard to tell (both improves in fear, but decrease in disgust)
 (6) (Accessaries) it seems that both methods (MaGNET-pixel and MaGNET-style) can improve the occurrence of headwear
But all the results are qualitative, the reviewer would like to see some quantitative results, such as how close the new distribution sampled using the MaGNET methods is to the uniform distribution, under the same categories (Gender, Hair, Glasses, Age, Emotion, Accessaries) provided by the authors comparing to the previous methods.

Minor comments:
1. What is the per-region slope matrices $\mathbf{A}_i = \mathbf{J_S}(\mathbf{z}_i)$ in the sampling procedure of the MaGNET?
2. The resolution of some figures (Figure 5, 8, 15 - 20) are too low, especially in Figure 15 - 20 where the quality of generated samples is hard to verify.



**Summary Of The Paper:**

This paper proposes a sampling method called MaGNET for generative models which aims at sampling data from the latent distribution of the images uniformly.

**Summary Of The Review:**

This paper does not provide enough experimental results (both qualitatively and quantitively) to support their claims.

---

> ### Author Response · Authors · 2021-11-22
> **Reply to Reviewer 5nSv Comments (Part 1)**
>
> We thank the reviewer for their helpful feedback and kind suggestions. Below, we present our response to individual questions raised by the reviewer.
>
> **Q1.How does the MaGNET sampling affect the quality of the generated images? Common GANs literature provides some quantitative evaluation metrics (inception score, FID score, KID score) as a justification for their proposed methods, but this submission does not provide any justification using the prevalent evaluation metrics.**
>
> A1. DGN metrics that use distribution distance measures to compare generation quality, e.g., FID, KID are not always optimal for the evaluation of MaGNET, because MaGNET explicitly changes the output distribution of a DGN from the distribution learned from training data. Therefore, we use Precision-Recall [1,2] as a method to compare the quality (precision) and diversity (recall) of generation.
>
> Nevertheless, we agree that metrics like FID are the gold standard to evaluate overall quality of generation. Therefore, we perform the following experiment to portray the utility of MaGNET in increasing sample diversity. For StyleGAN2-f on FFHQ, we measure the FID for standard sampling vs MaGNET sampling, at different truncation values. We vary truncation since truncation increases the quality of generation while reducing diversity [3,4]; whereas MaGNET increases the diversity via uniform sampling, while possibly reducing quality. We see that for latent space truncation $\psi$ of {0.7,0.5}, which are presented by Karras et al. for greater generation quality [4,5], MaGNET improves (reduces) FID by 1.93 and 3.86 points respectively. For state-of-the-art truncation ($\psi=1$, therefore no truncation) in terms of FID, we introduce a proportion of MaGNET samples to compromise between increasing diversity and reducing quality. Doing so, we are able to improve the state-of-the-art FID for StyleGAN2 using around 4% of MaGNET samples. We summarize the results in the following table, a full version of which is added to Appendix F. In all experiments we use 50K generated samples and all 70K FFHQ samples, which is the current standard for FID calculation [6,7].
>
> | Truncation $\psi$ | Method | FID |
> |----|-----|-----|
> |1 | StyleGAN2$^{* \dagger}$ [5] | 2.84 |
> |1 | StyleGAN3$^{\dagger}$ [6] | 2.79 |
> |1 |   StyleGAN2                                   |    2.74    |
> |1 |   StyleGAN2  w\ 4% MaGNET $^{\dagger \dagger}$       |    **2.66**    |
> |.9 |   StyleGAN2                                   |    5.05    |
> | .9 |   StyleGAN2 w\ 20% MaGNET      |    **4.29**    |
> |.7 |   StyleGAN2                                   |   21.34   |
> | .7 |   StyleGAN2 w\ 100% MaGNET    |   **19.41**   |
> |.5 |   StyleGAN2                                   |   58.33   |
> |.5 |   StyleGAN2 w\ 100% MaGNET    |   **54.47**   |
>
> $^*$evaluated using 50K training samples
>
> $\dagger$ Paper reported metrics
>
> $\dagger\dagger$ New state-of-the-art for FFHQ 1024x1024 FID
>
> We have also added a new figure, Fig.4-right, where we show the evolution of FID for $\psi=1$ while progressively adding more MaGNET samples for FID evaluation. While these results show that there is clear utility of MaGNET in image generation, we would like to highlight that depending on the application, the significant increase in diversity offered by MaGNET could be more beneficial than the loss in quality.
>
> **Q2. The provided samples of the generated images (Figure 15 - 20) are not clear enough to provide a good justification for their visual quality**
>
> A2. Due to suboptimal compression prior to submission, the quality of some figures got significantly reduced. We have fixed the compression issue in our revised manuscript and apologize for the low-quality figures.
>
> ----
>
>
> [1] Sajjadi, Mehdi SM, et al. "Assessing generative models via precision and recall." NeurIPS 2018.
>
> [2] Kynkäänniemi, Tuomas, et al. "Improved precision and recall metric for assessing generative models." NeurIPS 2019.
>
> [3] Brock, Andrew, Jeff Donahue, and Karen Simonyan. "Large scale GAN training for high fidelity natural image synthesis." ICLR 2019.
>
> [4] Karras, Tero, Samuli Laine, and Timo Aila. "A style-based generator architecture for generative adversarial networks." CVPR 2019.
>
> [5] Karras, Tero, et al. "Analyzing and improving the image quality of stylegan." CVPR 2020.
>
> [6] Karras, Tero, et al. "Alias-free generative adversarial networks." NeurIPS 2021.

---

> ### Author Response · Authors · 2021-11-22
> **Reply to Reviewer 5nSv Comments (Part 2)**
>
> **Q3. It is hard to tell whether the proposed method truly improves the uniformity of the sampled data (in Figure 22)**
>
> A3. We thank the reviewer for taking note of the changes in attribute distributions, and we agree with the reviewer that uniform sampling on the manifold might not guarantee uniformity in semantic attributes. While we have clarified the difference between uniform on the manifold and uniform in semantic attributes in Sec 4.4 paragraph 2, we agree that we need to better differentiate uniform in distribution over the DGN manifold, and uniform in representation of attributes/semantic. We have now explicitly stated those differences in the introduction and abstract (e.g., in the abstract *“As uniform distribution does not imply uniform semantic distribution,  we also explore separately how semantic attributes of generated samples vary under MaGNET sampling”*) to reinforce the point that was originally only made in Sec 4.4 paragraph 2 and in the conclusion. What clearly transpires from the figures, however, is that MaGNET does considerably increase the visual diversity of the samples. We would like to reiterate that the key contribution/claim of our paper is attaining uniform distribution on the learned data manifold, not uniformity of semantic attributes.
>
> **Q4. What is the per-region slope matrices**
>
> A3. CPA DGNs (see Equation 1) introduce a partitioning of the latent space. Each input (latent vector) is mapped to the output space via an affine operation, the slopes and offset parameters of which depend on the region in which the given input falls into. We refer to these as the per-region slope and offset parameters. We have added references to Equation 1 in the main part of the paper when referring to per-region slope matrices
>
> **Q5. What is $A_i=J_S(z_i)$ in the sampling procedure of the MaGNET**
>
> A5. Thanks to the CPA form of the DGN (Eq. 1), $A_i$ corresponds exactly to the Jacobian matrix of the DGN $S$ at $z_i$. This can be obtained via automatic differentiation on most current deep learning libraries. We have clarified that point right after the itemized pseudo-code in Section 3.2. To further improve clarity we also added a pseudo-code in the Appendix G and referred to it in main. We also went through the paper to ensure that all notations are properly introduced.

---

> > ### Comment · Reviewer_5nSv · 2021-11-22
> > **Thanks for the replies and score has been updated.**
> >
> > The reviewer thanks the authors for the replies, the replies clearly address all of the concerns, the score is updated accordingly.

---

### Official Review · Reviewer_TJkz · 2021-11-03

**Correctness:** 3
**Technical Novelty And Significance:** 4
**Empirical Novelty And Significance:** 2
**Recommendation:** 5
**Confidence:** 4

**Main Review:**

Applying the change of variables formula to augment sampling from DGNs is a novel idea. Moreover, the method is interesting and theoretically well-motivated. Finally, the sampling algorithm itself is very straightforward.

However, I take issue with the framing of the algorithm in the abstract and introduction. Namely, the authors use the colloquial understanding of “uniform” side-by-side with the differential geometric / measure theoretic understanding of “uniform”. For example, in the abstract, the authors state that 1) many generative models today are trained on non-uniform data, which has “potential implications for fairness, data augmentation, anomaly detection, domain adaptation, and beyond,” and 2) their algorithm “produces a uniform distribution on the manifold regardless of the training set distribution”. This creates the false impression that the present technique is capable of neutralizing the negative "implications" on fairness, data augmentation, etc etc. This juxtaposition may imply parity between the two definitions of "uniform" to the inattentive reader. While the authors do emphasize the difference between the term in these two contexts much later in the paper, I feel that it is not appropriate for the abstract to mislead in this manner. Again, I only have this issue with the framing of the technique, not the technique itself.

WIth regard to the uniform sampling property of MaGNET, I have two concerns about the practicality of the method.
1. The authors have touched upon this, but uniform sampling from the data manifold does not imply uniform sampling of attributes. This is exacerbated when the model has not fully learned the manifold. Therefore MaGNET’s sampling is only as “uniform” as the DGN and the data manifold itself is.
2. Since the DGN can only be trained on the training data distribution, sample quality will vary across the true data manifold. Namely, sample quality will likely correlate with density w.r.t. the training data distribution. Therefore, I imagine that sampling uniformly will reduce sample quality overall. This seems to be corroborated by qualitative comparison of original v. MaGNET samples in the paper figures.

Computationally, the authors demonstrate in Appendix D that sampling with $N$ past 250k does not affect the Precision-Recall metric, but I could not find what $N$ is in the experiments shown. And since each image sample requires computing the Jacobian of the DGN w.r.t. its input, I wonder what is the approximate computation time needed to sample $N=250,000$ times for each of the models.


**Summary Of The Paper:**

The authors propose a uniform sampling technique for deep generative networks (DGNs) inspired by the probabilistic change of variables formula. The technique works with any already trained DGN and does not involve any further training. (Though it does require back propagation w.r.t. the input $x$.) In essence, the algorithm works by drawing many samples N >> K from the DGN, then sampling from these $N$ samples with probability inversely related to their pushforward density (as computed by the change-of-variables formula).


**Summary Of The Review:**

This method itself is novel and interesting, and warrants acceptance into the conference. However, the current wording of the title and abstract can be misleading, and should be edited to remove confusion.

Pros:
+ Theoretically motivated
+ Algorithmically simple

Cons:
- Seems computationally expensive
- Oversells the capabilities of the technique in the abstract, namely: mathematically uniform is implied to mean semantically uniform
- Needs proofreading

---

> ### Author Response · Authors · 2021-11-22
> **Reply to Reviewer TJkz Comments (Part 1)**
>
> We thank the reviewer for taking note of the novelty of our method and their kind suggestions. Below, we present our response to individual questions raised by the reviewer
>
> **Q1. Though it does require back propagation**
>
> We thank the reviewer for highlighting this required property for MaGNET to be applied. We have added a remark in the introduction (second paragraph) right where we introduce MaGNET. We assume that the DGN outputs are differentiable w.r.t. their inputs, as most DGNs are usually trained through gradient descent (and are thus differentiable).
>
> **Q2. Framing of the algorithm in the abstract and introduction using “uniform” side-by-side with the differential geometric / measure theoretic understanding of “uniform”**
>
> We agree that we needed to better differentiate uniform in distribution over the DGN manifold, and uniform in representation of attributes/semantics. We have now explicitly stated those differences in the introduction and abstract  (e.g. in the abstract “As uniform distribution does not imply uniform semantic distribution,  we also explore separately how semantic attributes of generated samples vary under MaGNET sampling”) to reinforce our original discussion from Sec 4.4 paragraph 2 and the conclusion.
>
> **Q3. Relation between DGN training distribution and manifold approximation leading to reduced quality for MAGNET sampling overall.**
>
> We thank the reviewer for raising this important point. It is true that the approximation of the data manifold is very likely to be correlated to the training set density on the manifold.  Therefore, using MaGNET sampling will generate lower quality samples at the regions of the data space where the DGN approximation is poor. To quantify the effect of MaGNET in generation quality, we have performed additional experiments. For StyleGAN2-f on FFHQ, we measure the FID for standard sampling vs MaGNET sampling, at different truncation values. We vary truncation since truncation increases the quality of generation while reducing diversity [1,2]; whereas MaGNET increases the diversity via uniform sampling, while possibly reducing quality. We see that for latent space truncation $\psi$ of {0.7,0.5}, which are presented by Karras et al. for greater generation quality [2,3], MaGNET improves (reduces) FID by 1.93 and 3.86 points respectively. For state-of-the-art truncation ($\psi=1$, therefore no truncation) in terms of FID, we introduce a proportion of MaGNET samples to compromise between increasing diversity and reducing quality. Doing so, we are able to improve the state-of-the-art FID for StyleGAN2 using around 4% of MaGNET samples. We summarize the results in the following table, a full version of which is added to Appendix F. In all experiments we use 50K generated samples and all 70K FFHQ samples, which is the current standard for FID calculation [4].
>
> | Truncation $\psi$ | Method | FID |
> |----|-----|-----|
> |1 | StyleGAN2$^{* \dagger}$ [3] | 2.84 |
> |1 | StyleGAN3$^{\dagger}$ [4] | 2.79 |
> |1 |   StyleGAN2                                   |    2.74    |
> |1 |   StyleGAN2  w\ 4% MaGNET $^{\dagger \dagger}$       |    **2.66**    |
> |.9 |   StyleGAN2                                   |    5.05    |
> | .9 |   StyleGAN2 w\ 20% MaGNET      |    **4.29**    |
> |.7 |   StyleGAN2                                   |   21.34   |
> | .7 |   StyleGAN2 w\ 100% MaGNET    |   **19.41**   |
> |.5 |   StyleGAN2                                   |   58.33   |
> |.5 |   StyleGAN2 w\ 100% MaGNET    |   **54.47**   |
>
> $^*$evaluated using 50K training samples
>
> $\dagger$ Paper reported metrics
>
> $\dagger\dagger$ New state-of-the-art for FFHQ 1024x1024 FID
>
> We have also added a new figure, Fig.4-right, where we show the evolution of FID for $\psi=1$ while progressively adding more MaGNET samples for FID evaluation. While these results show that there is clear utility of MaGNET in image generation, we would like to highlight that depending on the application, the significant increase in diversity offered by MaGNET could be more beneficial than the loss in quality.
>
> -------
>
> [1] Brock, Andrew, Jeff Donahue, and Karen Simonyan. "Large scale GAN training for high fidelity natural image synthesis." ICLR 2019.
>
> [2] Karras, Tero, Samuli Laine, and Timo Aila. "A style-based generator architecture for generative adversarial networks." CVPR 2019.
>
> [3] Karras, Tero, et al. "Analyzing and improving the image quality of stylegan." CVPR 2020.
>
> [4] Karras, Tero, et al. "Alias-free generative adversarial networks." NeurIPS 2021.

---

> ### Author Response · Authors · 2021-11-22
> **Reply to Reviewer TJkz Comments (Part 2)**
>
> **Q4. Computationally, the authors demonstrate in Appendix D that sampling with N past 250k does not affect the Precision-Recall metric, but I could not find what N is in the experiments shown. And since each image sample requires computing the Jacobian of the DGN w.r.t. its input, I wonder what is the approximate computation time needed to sample N=250,000  times for each of the models.**
>
> N is the number of monte-carlo samples used to estimate the per-region affine parameters of the DGN (Eq. 1). We have added in Appendix D the computation times for individual models used in our experiments. We have also added an online algorithm based on rejection sampling, to allow rapid sampling once a reasonable number of per region affine parameters have been estimated (e.g. N=250,000), in Appendix G. Below, we provide the computation times for select models (see Appendix D for all) for a setup with Tensorflow 2.5, CUDA 11, Cudnn 8 and 1 NVIDIA Titan RTX.
>
> --------------------------
> Jacobian Calculation:
>
> -------------------------------
> StyleGAN2-e with random output projection of 256 dims- 5.03s/it
>
> BigGAN with random output projection of 256 dims- 5.95s/it
>
> ProgGAN with random output projection of 256 dims- 3.02s/it
>
> ---------------------
> Singular Value Decomposition:
>
> -----------------------------------------
> StyleGAN2-e .005s/it
>
> BigGAN .001s/it
>
> ProgGAN .004s/it
>
> ---------------------
>
> For StyleGAN2 for example, N=250,000 requires ~14 days to obtain. This only needs to be done once, and it is also possible to perform online sampling once it is calculated. The time required for this is relatively small compared to the training time required for only one set of hyperparameters, which is ~35 days and 11 hours [5]. We have added pseudocode for MaGNET sampling and online sampling in Appendix G.
>
> [5] https://github.com/NVlabs/stylegan2

---

### Official Review · Reviewer_orST · 2021-11-08

**Correctness:** 3
**Technical Novelty And Significance:** 4
**Empirical Novelty And Significance:** 3
**Recommendation:** 8
**Confidence:** 3

**Main Review:**

Strengths:
(a) Given any trained DGN, the paper gives a novel theoretical method to produce samples that are uniformly distributed on the learned manifold, regardless of the training set distribution. The approach is novel and solves the problem elegantly.
(b) It proves the proposed method for a mild assmption that DGN only comprises continuous piecewise affine (CPA) non-linearities, such as ReLu, absolute value, max-pooling.
(c) It gives convincing experiments on synthetic dataset showing that regardless of the training set distribution their MaGNET approach produces samples that are uniformly distributed.

Weakness: The paper needs improvement in writing.
(a) In section 3.2, the notation J_s(z_i) is used without explaining it. "compute the per-region slope matrices A_i = J_s(z_i)". Please define the notation and explain how to compute the slope matrices.
(b) A high level proof of the main theorem 2, in the main paper will help the reader understand the theorem better.
(c) the x-axis values of the two plots in Figure 3 are different by order of 100, it does not seem correct.

**Summary Of The Paper:**

This paper concerns with uniform sampling from deep generative networks such as GANs and VAEs. The training samples of DGNs are often biased as they are obatined based on preferences, costs, or convenience that leads to DGNs producing biased examples. This paper gives a gemoetry based sampler MaGNET, that given any trained DGN, produces samples that are uniformly distributed on the learned manifold. It theoretically proves, and empirically shows that the MaGNET produces a uniform distrbution on the manifold regardless of the training set distribution. The theoretical proofs require that the DGNs only comprise continuous piecewise affine (CPA) non-linearities, such as ReLu, absolute value, max-pooling. The three main contributions of the paper are as following:
(a) It characterizes the transformation incurred by a density distribution when composed with a CPA mapping.
(b) It derives an analytical sampling strategy that allows to obtain a uniform distribution on a manifold that is continuous and piecewise affine.
(c) It provides multiple numerical experiments validating the gains of their proposed method MaGNET.


**Summary Of The Review:**

The paper provides a novel provable method to produce samples that are uniformly distributed on the learned manifold, regardless of the training set distribution. The method is also well proven empirically through numerous experiments. The proposed method help address the problem fairness in samples produced from DGNs trained on not-so-well-representative training datasets.

---

> ### Author Response · Authors · 2021-11-22
> **Reply to Reviewer orST Comments**
>
> We thank the reviewer for taking note of the novelty of our method and their kind suggestions. Below, we present our response to individual questions raised by the reviewer.
>
> **Q1. Paper needs improvement of writing**
>
> A1. We have proofread the paper thoroughly and made necessary changes to improve clarity.
>
> **Q2. In section 3.2 the notation $A_i=J_S(z_i)$ is used without explaining it.**
>
> A2. We thank the reviewer for addressing the lack of clarity in this notation. In practice, $A_i$ is the Jacobian $J$ matrix of the DGN $S$ at $z_i$. This can be computed easily using automatic differentiation on current deep learning libraries. We have clarified it right after the itemized pseudocode in Section 3.2. To further improve clarity, we have also added a pseudocode in Appendix G and have added references to it.
>
> **Q3. A high level proof of the main theorem 2, in the main paper will help the reader understand the theorem better.**
>
> A3. We have added a brief overview of the proof, which mainly leverages Theorem 1 (the analytical form of the distribution on the DGN manifold). Combining Theorem 1 with the proposed MaGNET prior, the DGN distribution in the output space becomes constant over the DGN manifold, i.e. uniform on the support.
>
> **Q4. The x-axis values of the two plots in Figure 3 are different by order of 100, it does not seem correct.**
>
> A4. Figure 3 presents the variation of density for generated samples on the data manifold. For each training sample from MNIST (point on the manifold), we consider the number of generated samples within an $\epsilon$-ball region as a measure of empirical density $\eta$ (we have renamed it as $\eta$, previously $x$) at that point of the manifold. Here, $\epsilon$ is the average 1-nearest neighbor distance of MNIST training samples. In Figure 3, we present the histogram of $\eta$ (x-axis) for 10000 different points of the MNIST data manifold. We can see that the range of $\eta$ variations is 40x more for standard sampling (left) compared to MaGNET sampling (right), confirming that the density variation is significantly higher for standard sampling as opposed to MaGNET. For example, from Fig.3-left we can see that some points on the MNIST data manifold have up to 400 generated samples within an $\epsilon$-ball neighborhood when employing standard sampling. Whereas, the maximum number of generated samples within an $\epsilon$-ball neighborhood for the same 10000 points, according to Fig.3-right, is 10; effectively validating uniformity of MaGNET NVAE. Qualitative examples in Fig 22, confirm that MaGNET NVAE samples are indeed on the MNIST manifold.

---

### Decision · Program_Chairs · 2022-01-20

**Decision:**

Accept (Poster)

**Comment:**

The paper proposes a simple method for uniform sampling from generative manifold using change of variables formula. The method works by first sampling a much larger number of samples (N) from uniform distribution in the latent space and then does sampling by replacement (using probability proportional to change in volume) to generate a smaller number of final samples (k << N) that are seen as approximately sampled from a uniform distribution from the generative manifold.

Reviewers had some questions/concerns about the confusing language in the abstract and introduction around the use of the term "uniform" which the authors have addressed satisfactorily. Authors have also provided results on quality (FID metric) of the generated samples as asked by the reviewers.

While the proposed method is rather simple, has high computational cost, and novelty is marginal (as noted by two of the reviewers), reviewers agree it is above the acceptance bar.